# Rotating neurons for all-analog implementation of cyclic reservoir computing

Xiangpeng Liang [1,2,5], Yanan Zhong [1,3,5], Jianshi Tang [1,4✉], Zhengwu Liu [1], Peng Yao[1], Keyang Sun[1], Qingtian Zhang [1,4], Bin Gao [1,4], Hadi Heidari [2✉], He Qian[1,4] & Huaqiang Wu [1,4✉]

Hardware implementation in resource-efficient reservoir computing is of great interest for neuromorphic engineering. Recently, various devices have been explored to implement hardware-based reservoirs. However, most studies were mainly focused on the reservoir layer, whereas an end-to-end reservoir architecture has yet to be developed. Here, we propose a versatile method for implementing cyclic reservoirs using rotating elements integrated with signal-driven dynamic neurons, whose equivalence to standard cyclic reservoir algorithm is mathematically proven. Simulations show that the rotating neuron reservoir achieves record-low errors in a nonlinear system approximation benchmark. Furthermore, a hardware prototype was developed for near-sensor computing, chaotic time-series prediction and handwriting classification. By integrating a memristor array as a fully-connected output layer, the all-analog reservoir computing system achieves 94.0% accuracy, while simulation shows >1000× lower system-level power than prior works. Therefore, our work demonstrates an elegant rotation-based architecture that explores hardware physics as computational resources for high-performance reservoir computing.

[1] School of Integrated Circuits, Beijing National Research Center for Information Science and Technology (BNRist), Tsinghua University, Beijing 100084, China. [2] Microelectronics Lab, James Watt School of Engineering, University of Glasgow, Glasgow G12 8QQ, UK. [3] Institute of Functional Nano & Soft Materials (FUNSOM), Jiangsu Key Laboratory for Carbon-Based Functional Materials & Devices, Soochow University, Suzhou, Jiangsu 215123, China. [4] Beijing Innovation Center for Future Chips (ICFC), Tsinghua University, Beijing 100084, China. [5] These authors contributed equally: Xiangpeng Liang, Yanan Zhong. ✉email: jtang@tsinghua.edu.cn; hadi.heidari@glasgow.ac.uk; wuhq@tsinghua.edu.cn

Reservoir computing is a bioinspired machine learning paradigm introduced in the early 21st century[1–3]. The randomly and recurrently connected nonlinear nodes in the reservoir layer provide efficient implementation platforms for recurrent neural networks with low training costs (Fig. 1a). In principle, the complex dynamics generated by the reservoir nonlinearly map the input data to spatiotemporal state patterns in a high-dimensional feature space, where the state vectors of different classes can be linearly separated[1,4]. Furthermore, reservoir computing is a powerful approach for processing temporal signals

**Fig. 1 Reservoir computing architectures. a** A conventional reservoir computing architecture with random connections. **b** A simplified version of a reservoir, also known as a cyclic reservoir. The randomly connected neurons are replaced with a ring structure. **c** Illustration of the working principle of the proposed rotating neuron reservoir (RNR) that can be physically implemented. The input weights are uniformly distributed in the range of [−1, 1], and a pre-neuron rotor sends the signal to different neuron channels at different time steps. After flowing through the dynamic neurons, the signal is sent to different state channels via another post-neuron rotor, and the final states are read out through a fully connected layer and used in training. **d** Sketch of the working principle for the case of three neurons, where **R** denotes the rotation matrix. The legend for all subfigures is provided at the bottom.

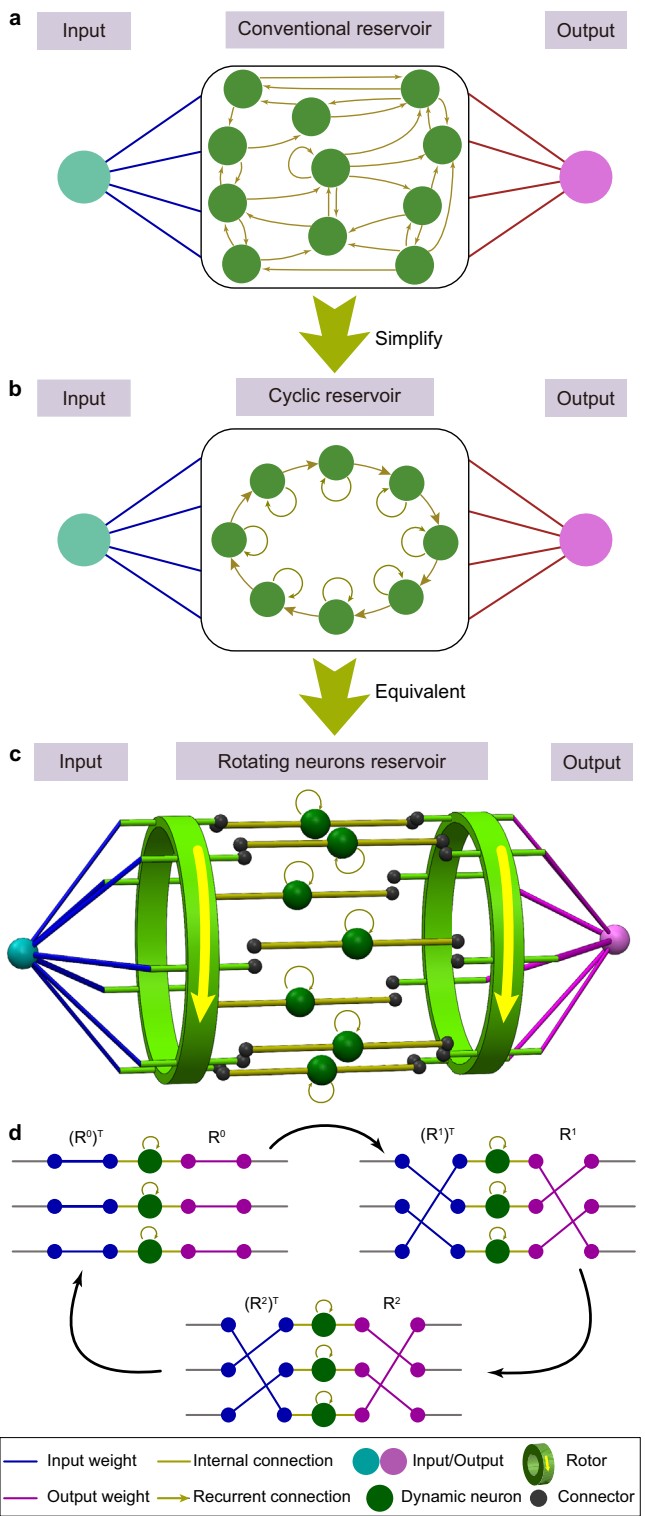

due to the recurrent connections that create dependencies between current and past neuron states, which is also known as short-term memory or fading memory[2,5]. In particular, reservoir computing has demonstrated excellent performance in complex time-series prediction and classification tasks[4,6].

Given the potential of reservoir computing, exploring physical dynamics as computational resources of reservoirs for highly efficient information processing has received considerable research attention in recent years. In 2011, a pioneer study[7] introduced a delay-based reservoir and the concept of virtual nodes into a physical implementation of a cyclic reservoir (CR), as shown in Fig. 1b which is a simplified reservoir without performance degradation[5]. This compelling finding provided an effective method for performing hardware-based reservoir computing, making it an attractive candidate in the field of neuromorphic computing. In follow-up studies, various emerging devices and systems were investigated as physical reservoirs[8], and they included spintronic devices[9], photonic devices[10–14], quantum devices[15], memristive devices[16–18], nanowire networks[19], and even soft robotic arms[20]. However, the main drawbacks associated with the use of delayed feedback and time-multiplexing are as follows: (i) delayed feedback is costly for hardware implementations using conventional complementary metal–oxide–semiconductor (CMOS) technology or optical approaches, which require additional digital components[7,21], such as analog-to-digital converters (ADCs) and random-access memory, or bulky optical fibers[10,11,22,23], respectively; (ii) in the absence of a delayed feedback line, a reservoir computing system cannot simultaneously maintain an appropriate memory capacity (MC) or satisfactory state richness. For example, previous research revealed that shortening the step size in time multiplexing could improve the MC but at the cost of reducing the state richness, or vice versa[16]. (iii) The serial operations in time multiplexing increase system complexity and latency for both input and readout, whereas parallel computing, which enhances the throughput, is more desirable in neuromorphic computing[24]. These obstacles hinder further reductions in power and size when the cost for an entire reservoir computer, from the signal input to the computing output, is considered; thus, a knowledge gap associated with massive deployment in practical applications remains. There is an urgent need to develop a new architecture involving hardware-based reservoir computers of miniature size with low power consumption and high capability for large-scale integration[8,25].

In this work, we propose a rotating neuron-based architecture for physically performing reservoir computing in a more intuitive way, namely rotating neurons reservoir (RNR), whose rotation behavior matches with the neurons update in a CR, as rigorously proven through mathematical derivations. Compared with the existing implementations in reservoir computing[17,19–21,23], the

RNR is hardware-friendly, resource-efficient, fully parallel, and explainable by standard CR. To verify the feasibility and potential of the RNR, an electrical RNR (eRNR) design based on CMOS circuits is introduced together with a simulator. Furthermore, a prototype eRNR composed of eight parallel reservoir circuits is built to perform analog near-sensor computing, and real-time Mackey–Glass time series prediction and real-time handwriting recognition are successfully performed in hardware experiments. To realize an all-analog reservoir computing system, the eRNR is further integrated with an analog memristor array that implements the fully connected output layer. Through the proposed noise-aware training method, the conductance variation of the memristor array is accommodated, and high classification accuracy of 94.0% is achieved for a handwritten vowel recognition task. Finally, a CMOS circuit simulation based on standard 65 nm technology indicated that the eRNR system is projected to consume as little as 32.7 μW of system power in the handwriting recognition task; this total would be more than three orders of magnitude lower than that achieved by literature-reported reservoir systems. These results highlight the tremendous potential of the proposed RNR, offering a promising paradigm for resource-efficient reservoir computers.

## Results

**Physical CR with rotating neurons**. The rotation couples the physical RNR and software CR. The mathematical derivation of the RNR proves that a rotating neuron array is equivalent to a CR model (Fig. 1b) as detailed in the Methods section. Figure 1c illustrates the operation principle of the rotation-based reservoir: if the neuron array is fixed, the pre- and post-neuron rotors rotate in the same direction to periodically shift the connections, which is equivalent to rotating the neuron while fixing the pre- and post-neuron rotors. Figure 1d shows an example of a three-neuron RNR. The rotors shift the connections before and after the neurons. The channels on the right-side output the analog computing results equivalent to the neuron states in a CR model with the same input. We shall mention that the fundamental of RNR is widely applicable to various rotating components, not limited to CMOS implementations, that can be developed as a reservoir by embedding dynamic neurons.

Thus, the main challenge of implementing a hardware RNR is the construction of the physical rotors and dynamic neurons based on the above approaches. Figure 2a illustrates a schematic of an $N$-neuron eRNR designed using CMOS circuits. The implementation of the input layer using binary weights is important because it allows the system to directly interface with analog sensory signals. $\mathbf{W_{in}}$ is taken to be a matrix consisting of a randomly generated uniform distribution of -1 and 1 values, which have been proven to be effective as multilevel weights[26]. Assuming that the signal source is $u(t)$, for each neuron, the driving signal should be $\gamma u(t)$ or $-\gamma u(t)$ during one-time step, where $\gamma$ is the input scaling factor. $\mathbf{W_{in}}$ can be configured by changing the switches ($S_1$ to $S_N$). Note that the $\mathbf{W_{in}}$ should remain unchanged while the RNR is operating so that the switches can be replaced with fixed connections.

Next, the pre-neuron rotor is implemented using $N$ $N$-channel multiplexers composed of transmission gates. All multiplexers share a common address line from a $\log_2 N$ (for $N = 2, 4, 8, 16 \ldots$) bit counter but different channel sequences for neuron connections, as illustrated in Fig. 2a. A driving clock with a period of $\tau_r$ is used to sequentially increase the counter address from 0 to $N - 1$ and then reset it to 0. This address is used to control the activated channels of all the multiplexers. Because the sequence of neuron connections is inconsistent, every multiplexer is connected to a different neuron during one $\tau_r$. Such a configuration ensures that

every input channel transmitting $\gamma u(t)$ or $-\gamma u(t)$ continues to poll every neuron during every rotation cycle $\tau_r \times N$, which corresponds to the transformation $\gamma (\mathbf{R}^{k-1})^{\mathrm{T}} \mathbf{W_{in}} \mathbf{u}(k)$ as described in the Methods section, where $\mathbf{R^{k-1}}$ denotes $(k\text{-}1)$-time-shifting. Upon receiving the neuron input $\gamma (\mathbf{R}^{k-1})^{\mathrm{T}} \mathbf{W_{in}} \mathbf{u}(k)$ and adding to its current value, the resulting neuron output $\mathbf{a}(k)$ is represented by the voltage level measured at the right side of the neuron circuit. The final step is to employ another post-neuron rotor at the output to convert $\mathbf{a}(k)$ to a state vector $\mathbf{s}(k)$. The post-neuron rotor performs an operation that is a mirror of that implemented by the input multiplexer array to obtain the forward rotation $\mathbf{R}$.

In addition to the rotors, dynamic neurons are also crucial elements in nonlinear computing. Based on the fundamental RNR characteristics described in the Methods section, a neuron in the RNR should possess three important characteristics: nonlinearity, integration ability, and leakage ability (Fig. 2b). Figure 2c illustrates a dynamic neuron specifically for the eRNR. Figure 2d and **e** plot the nonlinearity (a rectified linear unit (ReLU) that can be implemented with a diode) and integration characteristics (with a time constant $\tau_n = R_{int} \times C_{int}$ for the neuron), respectively. In the absence of the diode, the activation function becomes linear. The design and modeling of the dynamic neuron used in the eRNR are detailed in the Methods section. As discussed in Fig. 2b and the Methods section, most of the recently reported devices and materials for physical reservoir computing could also be used as the neuron in the RNR architecture[9,16,17]. Finally, an eRNR can be built by combining rotors and neurons. Multiple parallel RNRs can simultaneously connect to a common input signal but use different $\mathbf{W_{in}}$ configurations to increase the state richness. Figure 2f illustrates a complete eRNR computing architecture that includes $M$ parallel $N$-neuron eRNRs. The output weights are obtained through training and mapped in a memristor array to calculate the final results.

Moreover, a noise-free simulator was developed to evaluate the performance of the eRNR under different configurations and demonstrate its equivalence to a CR (as proven analytically in the Methods section). The first simulation was designed to confirm the consistency between the RNR and the CR and emphasize the role of rotation in the RNR. The key network characteristics based on different parameters, nonlinearities, and rotation directions were investigated. Before comparing the network characteristics of the software CR and the hardware RNR, a numerical method was developed to calculate the software CR parameters, such as the input scaling factor $\alpha$ and recurrent strength $\beta$, from the RNR behaviors to find the CR counterpart for a hardware RNR (see Methods). The prime task-independent network characteristic for a reservoir is the MC, which indicates its capability to retain the fading memory of the previous input[8,27] and plays a critical role in the reservoir's performance in temporal signal processing. The standard MC measurement is introduced in Supplementary Note 1. Figure 3a plots the MC as a function of reservoir size $N$ in different scenarios. We observed excellent agreement in the MC between the eRNR and its CR counterpart for both ReLU and linear activation functions. The ReLU neurons yielded a lower MC because the nonlinearity suppressed the fading information for previous inputs, as also observed in earlier studies[27,28]. For the RNR, we investigated the effect of the rotating direction to validate the design of the two rotors. The four lines at the bottom of Fig. 3a show the MC when the two rotors stopped or co-directionally rotated. The near-zero MC suggests that in cases with no rotation and counter-directional rotation, the RNR failed to implement reservoir computing functionalities since there was no MC for processing the temporal signal. In addition to MC, the other three important network characteristics are computing ability (CA), kernel quality (KQ), and generalization rank (GR)[29]

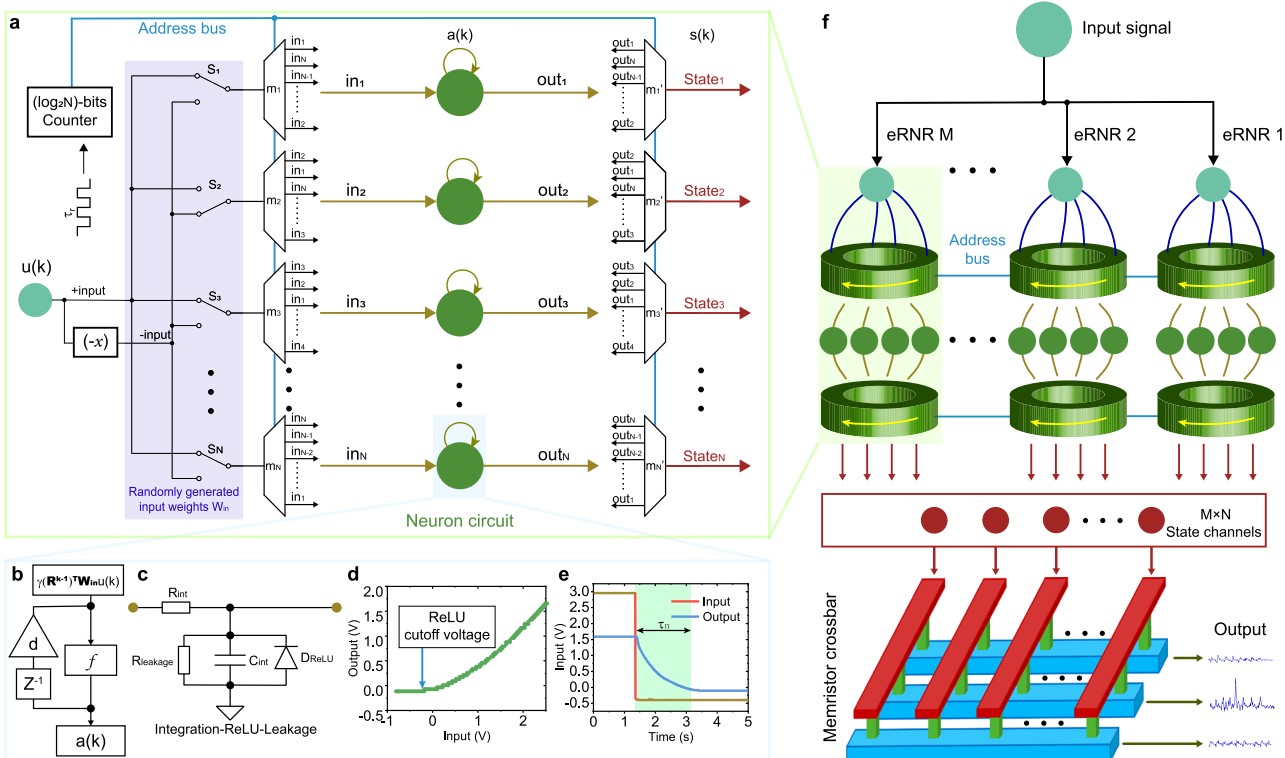

**Fig. 2 Implementation of the eRNR. a** Schematic of an $N$-neuron eRNR. Given an input $u(k)$, first, an operational amplifier generates another signal source $-u(k)$ or negative input. The switch array $S_1$ to $S_N$ determines the input weights $\mathbf{W_{in}}$ by selecting a positive or negative source for each multiplexer. The multiplexers $m_1$ to $m_N$ and $m_1'$ to $m_N'$ are involved in the electrical implementation of pre- and post-neuron rotors, respectively. The $\log_2 N$-bits counter outputs an address signal to sequentially activate the channels of each multiplexer at switch intervals $\tau_r$. Based on the distinct sequence of neuron connections ($in_1$ to $in_N$ for the input and $out_1$ to $out_N$ for the output), the behavior of the multiplexer array is equivalent to that of a rotor cyclically shifting connections between neurons and input/output channels. The sequence for output channels is a mirror version of that of input channels, which complies with the common-directional rotation principle in RNR theory. **b** General schematic of the dynamic properties required for a neuron in an RNR. When a neuron input $\gamma(\mathbf{R}^{k-1})^\mathsf{T}\mathbf{W_{in}}\mathbf{u}(k)$ that has been processed by a pre-neuron rotor and input weights are provided, the neuron performs nonlinear transform $f$, integration (feedback line), and leakage (decay factor $d$) operations on the signal. $a(k)$ is the neuron output at the $k$th step. **c** A dynamic neuron in the eRNR. $C_{int}$ and $R_{int}$ serve as integrators. The rectifying diode $D_{ReLU}$ provides an activation function similar to a nonlinear ReLU function. Finally, high resistance $R_{leakage}$ is added to control the current leakage rate, that is, the decay factor $d$ in Eq. (6). **d, e** The nonlinear properties (**d**) and dynamic integration (**e**) of the neuron for $R_{int} = 10$ kΩ, $C_{int} = 1$ μF, and $R_{leakage} = 100$ kΩ. $D_{ReLU}$ is a germanium diode with a forward voltage of approximately 0.3 V. **f** Schematic of a complete eRNR system that includes $M$ parallel $N$-neuron RNRs. The total length of the state matrix is $M \times N$. The voltage signal of each state channel is multiplied by the trained output weights stored in a memristor array to yield the final computing result.

(see Supplementary Note 1). These factors were analyzed by varying the time constant of neurons $\tau_n$, which also changed the parameter matching result for the CR counterpart. As shown in Fig. 3b, the network characteristics of the physical eRNR again matched that of its CR counterpart. Here, the minor difference may be attributed to the imperfect diode characteristics as a ReLU function. The results presented in Fig. 3a, b corroborate the finding that a properly configured RNR (rotation in a common direction) is equivalent to a software-based CR and hence can be used for implementing physical reservoir computing.

**The performance benchmark for the eRNR.** As an implementation of reservoir computing, the eRNR should be able to approximate a nonlinear system, for which a nonlinear autoregressive moving average system (NARMA) is a widely recognized benchmark for testing reservoir computing performance. A standard tenth-order NARMA system can be expressed by the following formula:

$$y(k+1) = 0.3y(k) + 0.05y(k)\sum_{i=0}^{9} y(k-i) + 1.5x(k)x(k-9) + 0.1 \quad (1)$$

where $x(k)$ is a randomly generated white noise input in the range

of [0, 0.5] and $y(k+1)$ is the target number. As can be observed in Eq. (1), the recursive configuration demands both nonlinear fitting and MC for the prediction model. In this task, an eRNR model was used to receive the $x(k)$ input and then predict the $y(k+1)$ output after training. In total, 4000 data samples ($x(k)$ and $y(k)$) for NARMA10 were generated to train (3000 samples) and test (1000 samples) the eRNR model. Given the same $x(k)$, the normalized root mean square error (NRMSE) of the predicted result $y'(k)$ versus $y(k)$ calculated with the NARMA10 model based on Eq. (1) was used to quantify modeling performance. In the first trial, two key parameters of the eRNR, the input scaling factor $\gamma$ and time constant of dynamic neurons $\tau_n$, were assessed while other parameters were fixed to obtain the optimal NRMSE for a single 400-neuron eRNR. The input scaling factor changes the effective range of nonlinearity, and the time constant affects the decay factor $d$. The noise-free simulation result is plotted in Fig. 3c, where the optimal value (NRMSE = 0.078) was found at $\gamma = 0.061$ and $\tau_n = 1.1$ s. It is worth mentioning that in a neuromorphic computing system, the electronic devices directly interacting with the environment and natural signals could exhibit a much longer time constant (e.g., greater than millisecond scale) compared with that of typical digital systems[30]. A

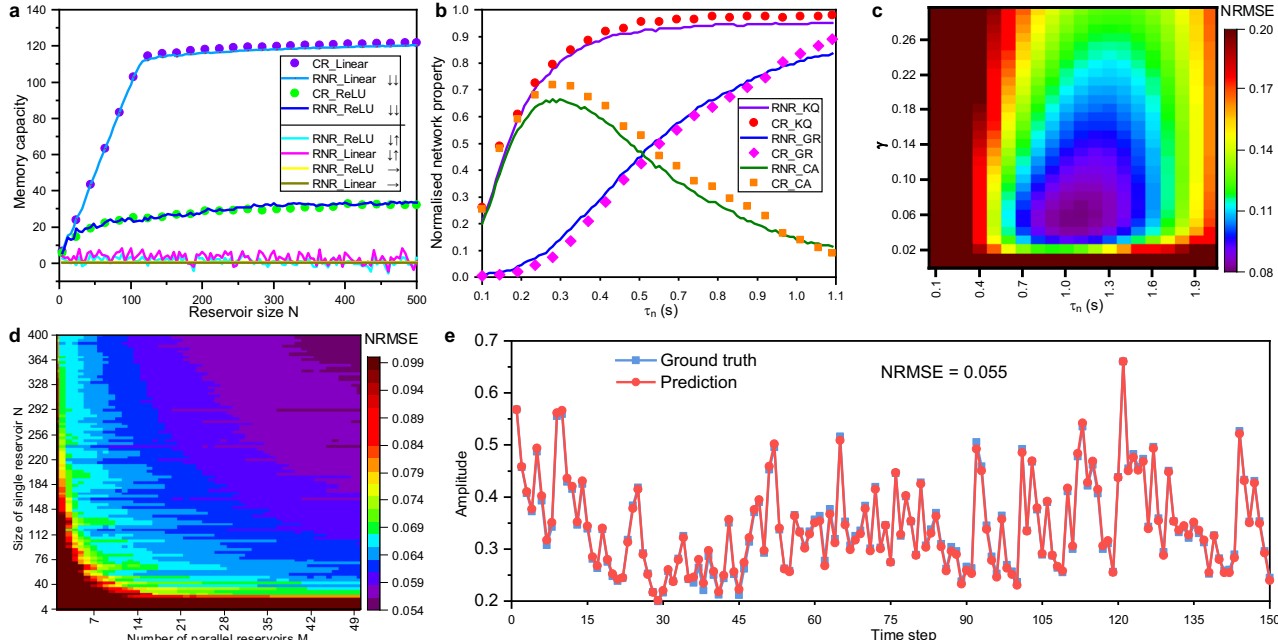

**Fig. 3 eRNR simulation results for network characteristics and nonlinear system approximation. a** Memory capacity (MC) versus the reservoir size $N$ for different scenarios. The two blue lines plot the MC of the eRNR using dynamic linear and ReLU neurons, respectively. The purple and green dots are obtained from the parameter-matched CR counterparts. The remaining four lines show the MCs of dysfunctional RNRs (counter-directional rotation and no rotation). The symbols '↓↓', '↓↑' and '→' indicate that the pre- and post-neuron rotors perform common-direction rotation, counter-directional rotation, and no rotation, respectively. The parameters are $\tau_n = 1$ s, $\tau_r = 0.125$ s, $\gamma = 0.5$, and $M = 1$. **b**. The computing ability (CA), generalization rank (GR), and kernel quality (KQ) as a function of $\tau_n$ for the dynamic neurons. For every $\tau_n$ value, the properties of the RNR are first calculated. Then, the CR counterpart is calculated through the parameter matching method, and the results are analyzed. The obtained parameters are $\tau_r = 0.125$ s, $\gamma = 0.5$, $N = 200$, and $M = 1$, and nonlinearity is provided by the diode. **c** NRMSE result for the NARMA10 system approximation task based on the two key parameters: the time constant $\tau_n$ and input scaling factor $\gamma$. The other parameters are $N = 400$ and $M = 1$. **d** NRMSE result for the NARMA10 modeling task when varying the reservoir size $N$ and the number of parallel reservoirs $M$. The parameters are $\tau_n = 1$ s, $\tau_r = 0.125$ s, and $\gamma = 0.05$. **e** An example prediction result $y'(k)$ and the ground truth $y(k)$ when NRMSE = 0.055, that is, for the best result obtained in (**d**). The parameters are $\tau_n = 1$ s, $\tau_r = 0.125$ s, $\gamma = 0.05$, $N = 388$, and $M = 50$ in this case, and a diode with a ReLU function is used.

fast time constant could result in an insufficient MC for retaining historical information. Such biologically realistic time constant values ($\tau_n$ and $\tau_r$, from milliseconds to seconds scale) were used throughout the explored hardware implementation and simulation processes. The performance can be further improved by increasing the number of parallel reservoirs $M$ with different input weights $\mathbf{W_{in}}$ as illustrated in Fig. 2f. As shown in Fig. 3d, the resulting NRMSE can be clearly reduced by increasing $M$ or $N$. The minimum NRMSE achieved in this experiment is 0.055 at $N = 388$ and $M = 50$. Figure 3e shows an instance of the predicted value $y'(t)$ in comparison with the ground truth $y(t)$ when NRMSE = 0.055. To the best of our knowledge, the NRMSE values for both the single eRNR (0.078) and parallel eRNRs (0.055) are lower than those reported in the previous studies[7,31] in the field of reservoir computing. Notably, the exponential form of nonlinearity in the transition region of the diode (different from the ideal ON/OFF form in the ReLU function used by the software) enhances the state representation of the NARMA10 system. This result demonstrates the tremendous potential of the eRNR in high-order nonlinear system approximation due to the rich physical dynamics of electronics devices.

**Physical eRNR implementation: real-time chaotic signal prediction**. The eRNR design can be implemented using commercial off-the-shelf components. Here, we developed a proof-of-concept prototype with $\tau_n = 1$ s, $N = 8$, and $M = 8$, as shown in Fig. 4a. The eight parallel eRNRs shared common power, counter, positive input, and negative input characteristics. The input weight

$\mathbf{W_{in}}$ varied for every eRNR to create diverse neuron dynamics and increase the state richness. More details about the prototype can be found in Supplementary Note 2. To evaluate the state generation performance, the first experiment with the $8 \times 8$ eRNR system was a multistep ahead prediction for Mackey–Glass chaotic system, which has been used in various reservoir computing studies as a benchmark task[1,17,32]. The Mackey–Glass system is defined by

$$\frac{dy}{dt} = \beta \frac{y(t - \tau)}{1 + y(t - \tau)^n} - \gamma y(t) \quad (2)$$

where the system parameters $\gamma$, $\beta$, and $n$ were set to the widely used values 0.1, 0.2, and 10, respectively. Additionally, the system is chaotic when $\tau > 16.8$, and predictions become correspondingly more difficult. In this experiment, we set $\tau = 17$ and the initial value $y(0) = 1.2$ following previous works. The samples generated based on the Mackey–Glass system were input into the $8 \times 8$ eRNR system with a sampling rate of 8 Hz. This sampling rate should be the same as the driving frequency of the counter to ensure that every sample point is captured; that is, $\tau_r = 0.125$ s. Based on this configuration, the 64 parallel output channels produce state values of the measured voltage for postprocessing. With our customized demonstration platform (the description of this platform is available in Supplementary Note 2), the Mackey–Glass chaotic signal $y(k)$ was continuously fed into the eRNR system. The training state matrix $\mathbf{s(k)}$ with a length of 64 based on $y(k)$ was used for output weight $\mathbf{W_{out}}$ training through linear regression, and the target value was input into the Mackey–Glass dataset shifted by $i$ steps ($y(k + i)$). Here, the

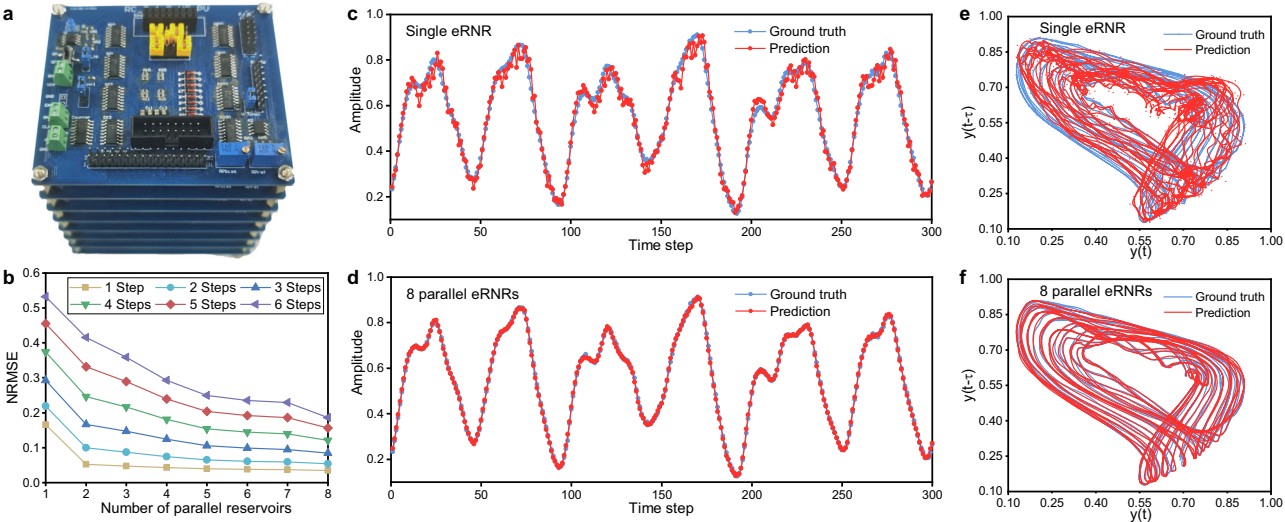

**Fig. 4 8 × 8 eRNR prototypes for Mackey–Glass time series prediction. a** An eRNR prototype consisting of eight 8-neuron eRNRs (i.e., $M = 8$ and $N = 8$). **b** NRMSE result for multistep-ahead Mackey-Glass time series prediction. The state matrix used in this experiment was obtained from the parallel output channels of the eRNR hardware. **c**, **d**. Two cases of one-step-ahead prediction with the Mackey–Glass time series result compared with the ground truth using **c** one eRNR (NRMSE = 0.17) and **d** eight parallel eRNRs (NRMSE = 0.03). **e**, **f** Phase space of the prediction compared with the ground truth using (**c**) one eRNR and (**d**) eight parallel eRNRs. The phase diagram was created by plotting the predicted and ground truth series $y(t)$ for the x-axis and $y(t − \tau)$ for the y-axis.

number of shifted steps $i$ depended on how many steps ahead of $y(k)$ the system could predict. The system continuously received $y(k)$ without any preprocessing and produced 64 state outputs, which were multiplied by $\mathbf{W_{out}}$ to predict the value $y'(k + i)$. This process was performed in real-time with the demonstration platform, and all the data, including $y(k)$, $y'(k + i)$, and $\mathbf{s(k)}$, were visualized (see Supplementary Movie 1).

To better understand how the number of parallel RNRs (i.e., M) affected the prediction performance of the system, the states within 360 s ($2880 × 64$ samples, half for training and half for testing) were collected with the platform. Again, the NRMSE was used to quantify the difference between the actual values $y(k + i)$ and the predicted values $y'(k + i)$. The result is shown in Fig. 4b. As $i$ increased, the time series became increasingly difficult to predict, resulting in a higher NRMSE; however, this NRMSE increase can be alleviated by using additional parallel reservoirs to enhance computational performance. Two examples of one-step-ahead prediction using one reservoir (NRMSE = 0.17) and eight parallel reservoirs (NRMSE = 0.03) are plotted in Fig. 4c, d, respectively. The traces of $y(k + i)$ and $y'(k + i)$ in the phase space were also examined (Fig. 4e, f). The traces of eight eRNRs exhibited excellent consistency with the true values compared with the traces for the one-reservoir system. These experimental results suggest that the $8 × 8$ eRNR prototypes can be used to make accurate predictions of variables in the Mackey-Glass chaotic system after training. Even with the inevitable noise introduced by the analog circuits, the eRNR can successfully emulate the chaotic system, with a low NRMSE of 0.03. Moreover, our experiment revealed that the eRNR prototype can properly predict one-step-ahead for more chaotic signals ($\tau > 17$) (Supplementary Fig. 1a–f). In comparison, the system performance could degrade as $\tau$ increases in multistep-ahead prediction (Supplementary Fig. 1g).

**Demonstration of near-sensor computing: handwriting recognition.** In the literature, some previously reported reservoir computing demonstrations achieved relatively low power consumption for certain parts inside systems using emerging devices

and materials[9,16,17]. However, the operations for entire systems are usually overlooked. An interface between a sensory signal and the reservoir input is usually necessary, and assistive techniques, such as converting between digital and analog data, memory buffering, preprocessing and feature extraction, are also often required[7,9,17]. These sophisticated operations increase system complexity and power consumption but are necessary for conventional physical reservoir computing and remain a key challenge for practical deployment[8]. In this work, a prime advantage of our eRNR prototype is that it can directly receive analog sensory signals and produce the parallel state output without any digital memory use or preprocessing, which could considerably reduce the power consumption of the overall system. In fact, this strength is highly attractive for emerging applications in analog near-sensor computing; notably, the processor can act as a direct interface for sensory signals for cognitive computing purposes[33].

To demonstrate analog near-sensor computing, a resistive touch screen was employed to provide an analog sensory signal for a handwritten vowel recognition task. In the experimental setup, a front-end circuit converted the resistive variations into two continuous signals representing the $X$ and $Y$ coordinates of the activated pixel on the screen. The $8 × 8$ eRNR system used in the Mackey–Glass task was divided into two $4 × 8$ eRNR subsystems (i.e., $N = 8$ and $M = 4$) to process $X$ and $Y$ temporal signals, and the total length of the state channel was still 64. In this case, the two subsystems still shared common power and counter but had different positive and negative inputs from the $X$- and $Y$-axes. A photograph of the hardware is shown in Fig. 5a. This experiment demonstrates that five different handwritten vowels (A, E, I, O, and U) can be distinguished after high-dimensional nonlinear mapping in the eRNR. Additionally, one important advantage of using reservoir computing systems is that their short-term memory property allows the network to retain the fading information of previous inputs in the state matrix at each time step. Thus, the state matrix obtained at the end of a handwritten event contains the information for the entire handwritten trace. After training, the eRNR system can perform point-by-point analog reservoir state generation without

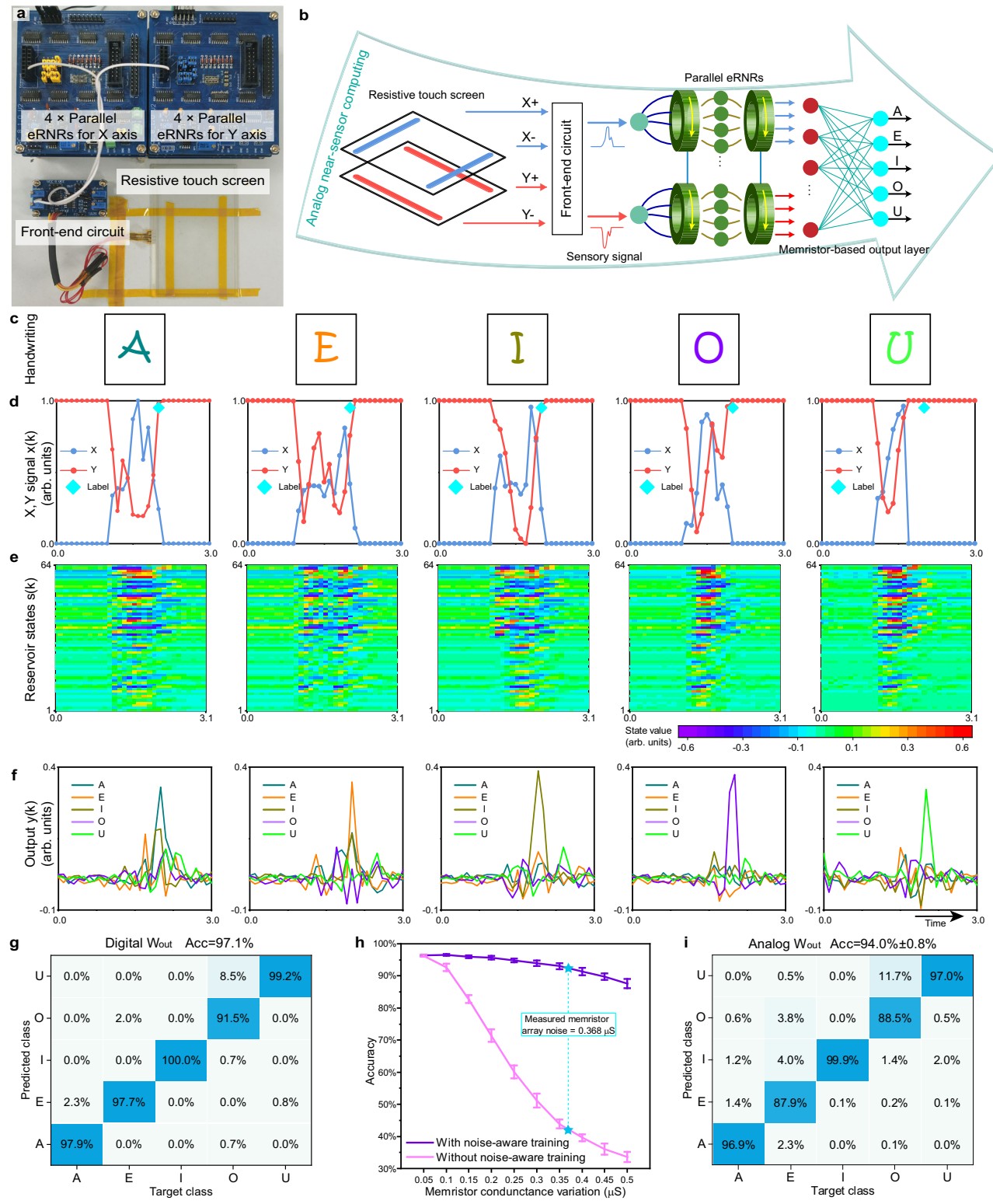

accessing digital memory. Consequently, the memory unit for storing a certain length of data, such as the data in a sliding window or segmented signal, in conventional machine learning approaches can be eliminated by making full use of the MC. Further advancement of this system involves the analog output weights stored in a memristor crossbar array to realize all-analog signal processing[34,35], for which the power consumption can be further reduced by taking advantage of the computing-in-

memory capability of memristors. Thus, from the sensory signal to the classification result, the entire system can perform near-sensor computing in the analog domain, as shown in Fig. 5b.

In our experiment, handwritten vowel data from eight participants were collected (see Methods), and typical handwritings are displayed in Fig. 5c. For different handwritten vowels, Fig. 5d shows the $X$ and $Y$ signals input into the eRNRs, and Fig. 5e shows the resulting state output of the 64 channels.

**Fig. 5 Analog near-sensor computing for handwriting recognition. a** The hardware used in this experiment: a handwriting sensor (resistive touch screen), a front-end circuit, and two 4 × 8 eRNR circuits for the x- and y-axes of the sensor. **b** A conceptual schematic of analog near-sensor computing without any digital memory. The front-end circuit drives the resistive touch screen and allows it to collect the handwriting information, which is then converted into two-dimensional x- and y-analog signals. These signals are then input into two 4 × 8 parallel eRNR circuits. The trained analog weights $\mathbf{W_{out}}$ in the memristor array are used to obtain the classification output for the five handwritten vowels. **c–f** The signal flows measured from the eRNR hardware for different handwritten patterns, including **c** the five handwritten vowels, **d** the sensory signals for the x- and y-axes $\mathbf{x(k)}$, **e** the 64 channel reservoir states $\mathbf{s(k)}$ of the eRNRs, and (**f**) the output $\mathbf{y(k)}$ computed based on $\mathbf{s(k)}$ and the trained weights. **g** Confusion matrix using $\mathbf{W_{out}}$ without noise-aware training. The overall accuracy is 97.1%. **h** Classification accuracy as a function of simulated memristor conductance variation with and without the noise-aware training method. The measured average variation of the memristor array was 0.368 μS. **i** Confusion matrix using analog $\mathbf{W_{out}}$ stored in the memristor array. The overall accuracy was 94.0%, with a standard deviation of 0.8%.

Using the labeling, training, and testing procedure introduced in the Methods section, 683 handwritten vowels (of a total of 703 in the test set) were correctly recognized, yielding a high accuracy of 97.1%. Examples of the point-by-point outputs for the five classes are illustrated in Fig. 5f, and the confusion matrix is shown in Fig. 5g. The errors mainly occurred when predicting 'O', which was misclassified as 'U' in some cases since these two classes are associated with similar writing traces. Here the software-trained $\mathbf{W_{out}}$ was deployed with the demonstration platform to perform real-time near-sensor handwriting recognition (see Supplementary Movie 2).

The next experiment further integrated the eRNR system with a memristor crossbar array that served as the output layer. In this experiment, a differential pair of two memristors was used to represent one synaptic weight, so 640 memristors were used to represent all the weights in the above $\mathbf{W_{out}}$ (see Methods and Supplementary Fig. 2). It is noted that the analog weights in a memristor array usually suffer from conductance variation issues (e.g., read noise) due to the nonideal device characteristics, leading to certain performance degradation compared with the floating-point digital weights in software[35]. The next simulation evaluated the effect of memristor conductance noise on the classification performance of the system to establish a proper training scheme. Figure 5h shows the result of directly mapping $\mathbf{W_{out}}$ without noise-aware training; notably, the accuracy decreased significantly as the noise level increased. In our experiment, the intrinsic noise of the memristor was the dominant noise source in the all-analog system. To achieve high accuracy, we adopted a noise-aware training method to obtain a robust $\mathbf{W_{out}}$ in the presence of memristor conductance variation[36,37]. In the noise-aware training scheme, Gaussian white noise with a standard deviation of ±0.03 was added to the normalized training state data before regression, and the resulting accuracy is plotted in Fig. 5h. The comparisons between digital $\mathbf{W_{out}}$, target analog $\mathbf{W_{out}}$, and the average values of the measured $\mathbf{W_{out}}$ after mapping are visualized in Supplementary Fig. 3. Most of the weight values can be successfully mapped to the memristor array with acceptable device variation, and the standard deviation (target conductance minus measured conductance) is approximately 0.368 μS. Finally, the confusion matrix using analog $\mathbf{W_{out}}$ measured from the memristor array is shown in Fig. 5i. Using the noise-aware training method and the measured analog $\mathbf{W_{out}}$, the classification accuracy was improved from 29.2 ± 0.9% (without noise-aware training) to 94.0 ± 0.8% (with noise-aware training). The recognition result for each participant is summarized in Supplementary Fig. 4.

**System-level power estimation and benchmark testing**. The power consumption for the whole eRNR-based reservoir computing system can be divided into two parts: eRNR circuit consumption and memristor array consumption. For the eRNR circuit, an 8-neuron eRNR was designed and simulated using a standard 65 nm CMOS process based on the parameters used in

the handwriting recognition task. The power estimation process and simulation are described in the Methods, where the power of eRNR was estimated by the simulation of the CMOS circuit using the foundry-provided library. The result indicates that the eRNR method can reduce the system power consumption for the handwriting task and chaotic signal prediction to 32.7 μW. The simulation also suggests that the static power, mainly associated with the dynamic neurons and the leakage current of transistors, plays a dominant role when the processing rate ($1/\tau_r$) is lower than 100 kHz (for which the power consumption was estimated to be 79.1 μW). This striking advantage is associated with the unique all-analog computing capability of our eRNR-implemented reservoir computing system, which saves the energy for frequent data conversion between digital and analog domains. It should also be highlighted that our all-analog eRNR provides more than three orders of magnitude lower system-level power consumption compared with previous cutting-edge reservoir computing systems, whose power are in the ranges of 83 mW to 150 W using different implementation methods (see Supplementary Table 1)[10,38–40].

As we can see, in contrast to conventional digital systems, the electronics' intrinsic dynamics were fully explored as computational resources in the all-analog eRNR architecture. A complete rotation-based reservoir computing system can be implemented by designing pre- and post-neuron rotors and dynamic neurons; this approach uses highly simplified hardware and is endorsed by the CR theory. Additional discussion and comparison of the power efficiency of the eRNR can be found in Supplementary Note 3.

## Discussion

In summary, we developed a hardware-friendly RNR architecture for all-analog neuromorphic computing; the resulting structure represents a fundamentally different reservoir architecture than those used in conventional hardware implementations. The proposed RNR has been validated in theory, simulation, and experimental analyses. The theoretical analysis of RNR rigorously mapped the CR algorithm onto the physical rotation of dynamic neuron array, providing a solid foundation for hardware implementation. Such an RNR can be embedded into natural rotating components in various electronics, mechanical systems, or even nanorobotics and empower them with computing capability. In the simulation using the eRNR model, the NARMA10 prediction task was performed to benchmark the system with varying hyperparameters, and record-low NRMSE values of 0.078 for a single eRNR and 0.055 for parallel eRNRs were achieved. It was found that the additional nonlinearity provided by the hardware-based dynamic neurons enhanced system performance in the approximation of the NARMA10 system, thus highlighting the computing potential of the proposed RNR. Furthermore, an 8 × 8 eRNR prototype was developed based on RNR theory for near-sensor analog computing. The prototype successfully demonstrated multistep-

ahead prediction of chaotic time series, and eight parallel reservoirs were found to reduce the prediction NRMSE from 0.17 to 0.03 for the studied Mackey–Glass chaotic system. This experimental result further validates the computing capability of our eRNR prototype under different experimental configurations. By further integrating the eRNR with an analog memristor array as the fully connected output layer, an all-analog reservoir computing system was realized to perform handwriting recognition tasks. A noise-aware training method was used to accommodate the conductance variation of the memristor array and improved the classification accuracy to 94.0%. In the simulation of the eRNR circuit, the overall system power consumption was estimated to be as low as 32.7 μW for the handwriting tasks operating at 10 Hz ($\tau_r = 0.1$ s), reflecting an advantage of more than three orders of magnitude compared to the consumption reported for reservoir computing systems in the literature. Additionally, further power analysis suggested that the static power, mainly dissipated by the dynamic neurons, dominates the system at processing rates below 100 kHz, while the overall system power remains at a low level for high processing rates (>100 kHz) (see Supplementary Table 1). This result can be explained by the fact that most computations occur in the analog domain that only contribute to static power, which is a general advantage of analog neuromorphic computing. Dynamic power, mainly attributed to logic switches and memristor arrays, starts to dominate the system at processing rates higher than 100 kHz (see Supplementary Table 2). Further discussion on the low-power advantage of eRNRs can be found in Supplementary Note 3.

To further enhance the eRNR system capabilities when performing complex tasks, a useful approach is to increase the number of neurons (N) or the number of parallel eRNRs (M) to expand the network size. Furthermore, a deep eRNR, consisting of multiple eRNR cells in series, could enhance the classification performance for inputs of different classes. From a hardware perspective, dynamic neurons could be replaced by recently reported emerging devices (e.g., dynamic memristors[16,17] and spintronic devices[9]) to further reduce the system size and power consumption. Different configurations of neurons could be beneficial for enhancing state richness and improving system performance. In addition, the eRNR design can be miniaturized and monolithically integrated onto chips to reduce power requirements and promote ultrafast computing. It is also worth mentioning that various rotational hardware could be explored for constructing efficient pre- and post-neuron rotors, which are the key to implementing the RNR. Our work demonstrates that the RNR is well-suited for large-scale and high-speed neuromorphic computing systems and has tremendous potential for use in applications involving the Internet of Things and edge computing, among others.

## Methods

**Fundamentals of the RNR.** For a typical reservoir computing with an $m$-dimensional input, an $n$-dimensional output, and $N$ neurons (Fig. 1a), the input coefficients $\mathbf{W_{in}}$ ($m \times N$) and reservoir weights $\mathbf{W_{res}}$ ($N \times N$) are randomly generated[1]. The complex dynamics stemming from the massive and random connections in the reservoir layer aid in nonlinearly mapping the $m$-dimensional input to the $N$-dimensional feature space where different input classes can be linearly separated. For $n$ output classes, only the output weights $\mathbf{W_{out}}$ ($N \times n$) need to be trained by using linear regression, which is relatively efficient compared to other recurrent neural network methods[1,2,41]. Note that linear ridge regression is used for training throughout this work. The neuron dynamics in the reservoir layer play an important role in signal mapping based on the following equation:

$$s(k+1) = f\left[\alpha \mathbf{W_{in}} u(k+1) + \beta \mathbf{W_{res}} s(k)\right] \quad (3)$$

where $s(k)$ denotes the neuron state matrix with length $N$ at the $k$th time step, $\mathbf{u}(k)$ is the $m$-dimensional input, $\alpha$ and $\beta$ are the scaling factors for the input and recurrent weights, respectively, and f(x) is a nonlinear transform function. In

reservoir computing, the reservoir layer $\mathbf{W_{res}}$ can be designed in a deterministic manner rather than being based on random connections[5]. In this case, the $\mathbf{W_{res}}$ becomes a shifted identity matrix $\mathbf{R}$

$$\mathbf{W}_{res} = \mathbf{R} = \begin{bmatrix} 0 & 0 & \cdots & 0 & 0 & 1 \\ 1 & 0 & \ddots & 0 & 0 & 0 \\ 0 & 1 & \ddots & 0 & 0 & 0 \\ \vdots & \vdots & \ddots & \vdots & \vdots & \vdots \\ 0 & 0 & \ddots & 1 & 0 & 0 \\ 0 & 0 & \cdots & 0 & 1 & 0 \end{bmatrix} \quad (4)$$

As a result, $\mathbf{W_{res}}$ is significantly simplified, and the network topology becomes CR, as shown in Fig. 1b. Previous research concluded that CR could achieve comparable results to those of conventional reservoir computing[5]. Then, the matrix $\mathbf{R}$ corresponds to one-time shifting in a ring structure, and $\mathbf{R^k}$ indicates a $k$-time cyclic shift analogous to physically rotating an object. As illustrated in Fig. 1d, it is assumed that (i) the post- and pre-neuron rotors are described by $\mathbf{R}$ and its transpose matrix $\mathbf{R^T}$, respectively; (ii) $a(k)$ is the dynamic neuron output at the $k$th step; and (iii) $s_r(k)$ is the state matrix of the RNR at the $k$th step measured at the end of each rotor's channel (before the output weights). Considering the rotation of the neuron output, the state $s_r(k)$ updating formula can be written as

$$\mathbf{R}^{k-1} a(k) = s_r(k) \quad (5)$$

which indicates that, at the $k$th step, the state matrix $s_r(k)$ is obtained by rotating the neuron output $a(k)$ for $(k-1)$ times. Furthermore, the output of dynamic neurons is determined based on both an input shift and the previous states

$$a(k+1) = f_r[\gamma(\mathbf{R}^k)^T \mathbf{W_{in}} u(k+1) + d a(k)] \quad (6)$$

where $d$ denotes the decay factor resulting from the dynamic property of the neuron (see the next subsection in the Methods), $\gamma$ is the scaling factor for the input, and $f_r(x)$ is the nonlinear transform implemented by the dynamic neurons. Equation (6) describes the signal flow through the neurons. Given an input $\mathbf{u}(k)$, it is first multiplied by the input weights $\mathbf{W_{in}}$. After $k$ reverse rotations of the input connections, the signal is fed into the dynamic nonlinear neurons, which output $a(k+1)$. If both sides of Eq. (5) are multiplied by $\mathbf{R^k}$, we can obtain

$$\mathbf{R}^k a(k+1) = \mathbf{R}^k f_r \left[\gamma(\mathbf{R}^k)^T \mathbf{W_{in}} u(k+1) + d a(k)\right] \quad (7)$$

Using Eq. (5), Eq. (7) can be simplified as

$$s_r(k+1) = f_r\left[\gamma \mathbf{W_{in}} u(k+1) + d \mathbf{R} s_r(k)\right] \quad (8)$$

Here, the excellent consistency between Eq. (3) and Eq. (8) reveals that the proposed physical RNR architecture (Fig. 1c) is equivalent to a software CR. Thus, a rotating object with dynamic neurons can act as a reservoir computer without using extra control units, ADC or memory, which remarkably reduces the system complexity and power consumption compared with those in conventional hardware implementation (see Supplementary Note 3).

**Design and modeling of dynamic neurons.** By observing Eq. (3), it appears that a dynamic neuron for the proposed RNR should satisfy three important characteristics as shown in Fig. 2b: provide a nonlinear activation function f(x); support integration ability for the summation between the current input and previous state $a(k-1)$; and support leakage, as related to the decay factor $d$, to avoid saturation caused by the integration process. Any passive element that exhibits these three characteristics could essentially be used as a dynamic neuron in the RNR architecture by fine-tuning the time constants of neurons and rotors. A dynamic node working in a physical reservoir may suffer from device variation issues, which impact system performance. Previous studies have revealed that a certain degree of device variation may be beneficial to system performance by enhancing state richness[16,17], but determining how to precisely control device variability warrants future explorations.

In implementations using standard electronics (Fig. 2c), a ReLU-type nonlinear transform can be provided by a diode, and the resistor $R_{int}$ and capacitor $C_{int}$ can act as integrators. Leakage can be considered by connecting the system to the ground via a large resistance $R_{leakage}$. In the simulation, this neuron can be modeled as follows:

$$\dot{V}_o(t) = \frac{1}{R_{int} C_{int}} V_i(t) - \frac{R_{int} + R_{leakage}}{R_{int} R_{leakage} C_{int}} V_o(t) + \frac{1}{C_{int}} I_s(e^{-\frac{V_o(t)}{V_T}} - 1) \quad (9)$$

where $V_o(t)$ and $V_i(t)$ denote the input and output voltages, respectively. The saturation current $I_s$ and thermal voltage $V_T$ stem from the Shockley diode equation $I = I_s(e^{\frac{V_D}{V_T}} - 1)$. The typical values for germanium diodes $I_s = 25 \times 10^{-9}$ A and $V_T = 0.026$ V were used in the simulation. In the case of linear neurons, the last term $\frac{1}{C_{int}} I_s(e^{-\frac{V_o(t)}{V_T}} - 1)$ should be removed from Eq. (9).

In our simulation, Eq. (9) was solved in MATLAB/Simulink. The discrete neuron output in Eq. (3) becomes $a(k) = V_o(k\tau_r)$. The pre- and post-neuron rotors can be modeled by continuously shifting $\mathbf{W_{in}} u(k)$ and the neuron output $a(k)$.

Since $R_{leakage}$ is a large resistance, the time constant associated with this neuron is mainly determined by the integrator $\tau_n = R_{int}C_{int}$. For the rate of rotation $\tau_r$, we normally use an empirical value of $\tau_r = \tau_n/8$.

**Parameter matching.** It has been analytically proven that a physical RNR can perform the same functionality as a CR (Eq. (8)). Therefore, given a properly configured RNR, its CR counterpart should exist and exhibit similar network characteristics. Parameter matching provides a numerical method to determine the CR counterpart. The main difference between a hardware RNR and a software CR is associated with nonideal dynamic neurons, which result in different amplitude ranges for integration and nonlinearity. Therefore, the objective is to find the appropriate scaling coefficients for the software activation function to approximate the hardware neuron output under the same input $\mathbf{W_{in}}u(k)$. An arbitrary $u(k)$ was generated as an input to the RNR, and the neuron output $\mathbf{a(k)}$ was obtained. Assuming that this $\mathbf{a(k)}$ is generated by a software CR, a comparative neuron update vector can be defined

$$\boldsymbol{a_p}\left(\boldsymbol{k+1}, \boldsymbol{\alpha}, \boldsymbol{\beta}, \boldsymbol{V_c}\right) = ReLU(\beta\boldsymbol{a(k)} + \alpha\mathbf{W_{in}}u(k), V_c) \qquad (10)$$

where $\boldsymbol{a_p}$ is the neuron output sequence of recurrent factor $\alpha$, input scaling factor $\beta$, and the ReLU cutoff value $V_c$. For certain values of $\alpha$, $\beta$ and $V_c$, the CR should match the RNR if the resulting $\mathbf{a_p(k)}$ is close to $\mathbf{a(k)}$ for any $k$. Hence, the CR counterpart of an RNR can be found by matching the three parameters. First, $V_c$ is the threshold voltage of the diode, unlike that in the ideal ON/OFF case in the software ReLU function ($V_c = 0$). This value can be obtained based on the minimum value of the $\mathbf{a(k)}$ sequence. Second, $\alpha$ and $\beta$ are determined by searching the potential values and finding those that minimize the NRMSE between $\mathbf{a(k)}$ and $\mathbf{a_p(k)}$, which can be described as $\min_{\alpha,\beta} NRMSE\left(\boldsymbol{a(k)}, \boldsymbol{a_p}\left(\boldsymbol{k+1}, \alpha, \beta, V_c\right)\right)$. For example, for an RNR with $\tau_n = 1$ s, $\tau_r = 0.125$ s, and $\gamma = 0.5$, the matched CR parameters are $\alpha = 0.87$, $\beta = 0.12$, and $V_c = -0.18$, and the corresponding MC values are compared in Fig. 3a.

**Handwritten vowel recognition using an eRNR.** The parameters of the eRNR used in the handwritten vowel recognition task are $\tau_n = 1$ s, $\tau_r = 0.1$ s, $N = 8$, and $M = 4$ (for each X and Y channel). All data were collected with our customized platform. In total, 66 channel data streams, including the two-axis signals and signals from 64 reservoir state channels, were collected at each time step. During the data collection process, eight participants were asked to write the five vowels on a resistive touch screen, and repeat at least 20 times for each vowel. Data for 1103 handwritten vowels (2802 s) were successfully collected. The location and class of each handwritten vowel were labeled at the final rising/falling edge of the X and Y raw data. We labeled the end of each handwritten vowel (the blue square in Fig. 5d) where the state matrix at this time step contains the information of the handwritten trace because of MC. Specifically, the $64 \times 1$ state matrix collected at the time denoted by the green dot can be considered a feature vector for the corresponding handwritten trace.

After data collection and labeling, the database was divided into a training set (400 handwritten vowels; 1025.8 s) and a testing set (703 handwritten vowels; 1776.2 s). According to the point-by-point computation introduced above, the size of the training label matrix $\mathbf{Y_{train}}$ for the five classes should be a five-dimensional data stream in which only the locations of green squares are set to 1, and values of 0 are assigned at other points. For training $\mathbf{W_{out}}$ ($64 \times 5$), ridge regression with the target $= \mathbf{Y_{train}}$ (five-dimensional label for 1025.8 s) and variables $= \mathbf{S_{train}}$ (64-dimensional state vector for 1025.8 s) was used. Next, $\mathbf{W_{out}}$ was multiplied by the test state matrix ($\mathbf{Y_{test}}' = \mathbf{S_{test}} \times \mathbf{W_{out}}$) to obtain a five-dimensional output representing the possibility of five potential classes at each time step, which corresponded to the graphs in Fig. 5f. To quantify the classification accuracy, the predicted output for the testing set $\mathbf{Y_{test}}'$ was compared with the manually labeled locations $\mathbf{Y_{test}}$. For every location in a handwritten event, for example, $y_{test}(k)|_{k = k_x}$, the actual output was investigated to find the maximum value in the range of $y_{test}(\mathbf{k_x} - 7)$ to $y_{test}(\mathbf{k_x} + 3)$. The corresponding channel that output the maximum value was considered the predicted class.

**Memristor-based output layer.** Memristor-based analog computing has displayed excellent potential in neuromorphic computing. While the input and reservoir layer are generally established based on eRNR design, the output layer, which employs standard vector-matrix multiplication operations, can be effectively implemented by a memristor array for end-to-end all-analog computing[42,43]. The memristor array has a unit cell of one-transistor-one-resistor (1T1R). Each 1T1R consists of a resistive switching memristor with a material stack of $TiN/HfO_x/TaO_y/TiN$ connected to a Si transistor that is fabricated using a standard 130 nm Si CMOS process[44,45]. The description of the memristor array can be found in Supplementary Fig. 2. As described in the main text, we used 640 memristors in total to represent 320 weights in the output layer. The computation principles of memristor-based analog computing can be expressed as $\mathbf{I} = \mathbf{V} \times \mathbf{G} = \mathbf{V} \times (\mathbf{G_p} - \mathbf{G_n})$, where $\mathbf{G}$ represents the weight matrix $\mathbf{W}$, and $\mathbf{G_p}$ and $\mathbf{G_n}$ are the positive and negative conductance matrices, respectively. Furthermore, we use a standard write-with-verify scheme to map the weight matrix $\mathbf{W_{out}}$ to the conductance of the memristor array[34].

**Power estimation.** As shown in Fig. 2a, the neurons, as passive components, are driven by the negative and positive sensory signals, providing a power source $P_s$. Also, the energy consumed by the counter and transmission gates depends on not only the static power but also the rate of rotation $\tau_r$. The total power consumption $P$ of the system consisting of $M$ 8-neuron eRNRs (where the number of neurons $N$ is fixed at 8) can be expressed as

$$P = P_c + \left(P_s + P_t + \frac{E_c^{dyn} + E_t^{dyn}}{\tau_r}\right) \times M + \frac{E_m^{dyn}}{\tau_r} \qquad (11)$$

where $P_c$ and $P_t$ represent the static power of the counter and transmission gates, respectively, and $E_c^{dyn}$ and $E_t^{dyn}$ represent the dynamic energy dissipated in the transition region driven by the rate of rotation $1/\tau_r$. $E_m^{dyn}$ is the energy consumed in the output layer (memristor array) for one inference. The $M$ parallel eRNRs can share one counter, but the power for the other components increases with the number of parallel eRNRs $M$. For our application involving real-time handwritten signals, the operation period $\tau_r$ is relatively slow (0.1 s) to match the time scale of human operations.

The simulation result shows that $P_s = 3.27$ μW, $P_c = 0.93$ μW, and $P_t = 0.70$ μW, regardless of how fast the rotors are operating. Moreover, the energy-related to the rotation rate is $E_c^{dyn} = 0.31$ pJ and $E_t^{dyn} = 0.07$ pJ. For the memristor-based output layer, the power dissipated by the voltage buffer driving the memristor array and the memristor array itself is 144 and 0.8 μW, respectively. During every $\tau_r$, the only one-time inference is needed since all state channels are monotonously increased or decreased. The memristor array takes ~50 ns to respond to the state voltage. Therefore, the dynamic energy of the memristor array for every inference step is $E_m^{dyn} = (144 \text{ μW} + 0.8 \text{ μW}) \times 50 \text{ ns} \times 64 = 463.36$ pJ/class. The total power consumption of an $8 \times 8$ eRNR can then be calculated using Eq. (11). The simulated power breakdown at different frequencies is shown in Supplementary Table 2. Notably, this result also reveals that the power would not considerably increase at rates of rotation ($1/\tau_r$) below 100 kHz since static power dissipation dominates the system.

## Data availability
The source data for Figs. 2–5 are provided in separate Source Data files. Other data that support the findings of this study are available from the corresponding authors upon reasonable request. Source data are provided with this paper.

## Code availability
The code for the eRNR simulator and NARMA10 task is available at https://github.com/Tsinghua-LEMON-Lab/Rotating-neurons-reservoir / (https://doi.org/10.5281/zenodo.5909080). Other codes that support the findings of this study are available from the corresponding authors upon reasonable request.

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

## Acknowledgements

This work was in part supported by China's key research and development program 2021ZD0201205 (H.W.), Natural Science Foundation of China 91964104 (J.T.), 61974081 (J.T.), 62025111 (H.W.), 62104126 (Y.Z.), XPLORER Prize (H.W.), 92064001 (B.G.) and the UK EPSRC under grant EP/W522168/1 (H.H.). We thank Beijing IECUBE Technology Co., Ltd. for their generous support of the testing system.

## Author contributions

X.L., Y.Z., and J.T. conceived and designed the experiments. X.L. set up the simulation, hardware prototype and conducted the experiments. Z.L., X.L., P.Y., and K.S. contributed to the memristor array measurement and power analysis. X.L., Q.Z., B.G., and H.Q. contributed to the data analysis. X.L. and J.T. wrote the paper with inputs from H.H All authors discussed the results and commented on the paper. J.T. and H.W. supervised the project.

## Competing interests

The authors declare no competing interests.
