## [Peer Review File · Nature Communications]

REVIEWER COMMENTS

Reviewer #1 (Remarks to the Author):

This manuscript reports on a study demonstrating a novel type of reservoir computing based on a cyclic reservoir. The study is noteworthy in two aspects:

- (i) it is the first to demonstrate a hardware realisation of the concept of a cyclic reservoir, showing its equivalence with a rotating neuron reservoir device;
- (ii) the novel hardware prototype developed in this study demonstrates a low-power end-to-end analog system capable of online learning.

The work is highly original and has potential to significantly impact the field of neuromorphic computing. It is refreshing to see such a compelling demonstration of physical reservoir computing. In my opinion, the novelty of the work and quality of the results merit publication in NCOMMS. The supplementary videos are impressive. This notwithstanding, I do have some suggestions below for improving the manuscript.

Main feedback

Discussion: The discussion as currently presented is a summary of the results. I would like to see this rewritten so that the authors discuss their results, both qualitatively and quantitatively, in the context of other similar studies. e.g. how do their NARMA10 simulation results compare to those reported by Appeltant et al. (2011) for a single cyclic reservoir? What are the similarities/differences. Similarly, how do their experimental results for the Mackey-Glass forecasting task compare to other physical RC studies? Can the authors comment on the ability to predict more chaotic MG signals (e.g. $\tau \gg 17$)? So, instead of predicting more steps ahead, predict one step ahead, but for a more chaotic signal. See also this paper that shows how training on more chaotic MG signals can improve prediction of less chaotic signals (i.e. demonstration of transfer learning): doi: 10.1109/ICRC2020.2020.00007

On line 347-348: please cross-reference supp table 1. Also, please add clarification that the stated power (32.7 μ W) is for 10 Hz while the memristor array dominates at higher processing rates. Can the authors similarly compare to other memristor-based RC systems?

Lastly, the authors should mention prospects for future studies that expand the capabilities of their system. e.g. online classification of more than 5 classes.

Minor comments and suggested edits:

1. line 44: the LSM-based RC work of Wolfgang Maass should also be acknowledged (Maass et al. 2002).
2. line 61: please cite RC studies using memristive nanowire networks (e.g. Lilak et al. 2021 <https://doi.org/10.3389/fnano.2021.675792> and refs therein).
3. lines 117, 119 (and possibly elsewhere): will be  is
4. line 128: please cross-reference Methods when mentioning "fundamentals of RNR".
5. lines 141-143: needs rephrasing, e.g. The simulator was developed to evaluate the performance of the eRNR and demonstrate its equivalence to a CR (as shown analytically in Methods).
6. line 171: NARMA  NARMA10 (because lower orders aren't chaotic - in fact, I'm not entirely sure even NARMA10 is technically chaotic, although it's certainly highly nonlinear)
7. line 177: typo with NARMA10
8. line 184: can be  is

9. lines 192 -194: this strong statement needs to be backed up with references and discussed in the Discussion (as per my comment above).
10. line 194: non-ideality  nonlinearity
11. line 207: remove "it has been studied that"
12. lines 238-239: cite references to back up this statement
13. line 361: remove "or backpropagation"
14. line 378: remove "which is of great interest"
15. lines 479-480: remove "the static power" and remove repeat of $P_s=3.27\mu W$.
16. line 593: Fig. 1 caption, possibly try placing the legend at the bottom because it applies to each fig
17. line 616: Fig. 2b caption, mention $a(k)$ is neuron output at k -th step
18. line 626: Fig. 3 caption, mention these are simulation results
19. line 646: Fig. 4 caption, clarify to $M=8$ and $N=8$

Reviewer #2 (Remarks to the Author):

Report on Rotating neurons for all-analog implementation 1 of cyclic reservoir computing

In this work, the authors realize an analog electronic implementation of cyclic reservoirs.

On the plus side:

- cyclic reservoirs have to my knowledge not been implemented before (while reservoirs with delay lines, which are very similar, have been extensively studied).
- the method to implement cyclic reservoirs, using a rotating logic element, is novel and elegant.
- the system is fully analog, including an analog output layer. An interface with a tactile screen is presented.
- An estimate of the energy consumption of an integrated system using CMOS technology is presented.
- Performance on tasks is good.
- The paper is mostly well written and easy to follow.

On the minus side:

- The trick of using a rotating logic element is a trick, not an important conceptual advance.
- The cyclic reservoir is only a minor variation on delay reservoirs, which have been much studied.
- A convincing argument for using cyclic reservoirs with rotating elements, rather than delay reservoirs, is not presented.
- The analog output layer has been presented in previous work.

Overall my feeling is that, while this is elegant and well implemented work, it does not reach the threshold of impact and broad interest for publication in Nature Communications.

Major comments:

The prototype system is not presented in detail. Please provide enough details for another researcher to be able to reproduce the experiment.

The paper is not very clear (particularly in the abstract and introduction) on what is experimental, what is simulation, and what is theoretical calculation.

If the authors claim that, in the context of CMOS implementations, their rotating architecture is more efficient than other architectures. They should present evidence. For instance it is not clear why a delay system with a single nonlinear node (Ref 6 Appeltant et al) could not be as energy efficient.

Minor comments:

I found it quite confusing to use the acronym RC for reservoir computer, and CR for cyclic reservoir computer. Initially while reading the paper, I thought CR was a typo.

Introduction :

“Through our noise-aware training method, the conductance variation of memristor array was accommodated and a high classification accuracy of 94.0% was achieved.” Imprecise. Classification of what?

“Finally, the system benchmark indicated that the eRNR system consumed as low as 32.7 microW for the handwriting recognition task, which was more than three orders of magnitudes lower than literature-reported reservoir systems.” Isn't this a theoretical estimate. The sentence suggests it is an experimental result.

Performance benchmark of eRNR. « the eRNR should be able to approximate a nonlinear chaotic system, for which NARMA is a widely recognized benchmark task to test RC performance.” NARMA is a Nonlinear Auto Regressive Moving Average system, not a nonlinear chaotic system.

“It is worth mentioning that the biologically realistic time constant values (...) were used throughout our hardware implementation and simulation so as to interact with the environment in biological time scales.” Unclear. What is biological? Why is slow good? Is it the electronic neurons that should imitate biological neurons? Or the task (handwriting recognition) that must be on the time scale of humans?

“To the best of our knowledge, the NRMSE values for both single eRNR (0.078) and parallel eRNR (0.055) are the lowest compared with the previous studies in the field of RC.” Maybe add some references?

Eq. 2. The Mackey Glass equation has a power at the denominator. Eq. 2 seems wrong?

Demonstration of near-sensor computing: handwriting recognition .“Using the datasets collected from the eight participants, the noise of our memristor array associated with the mapping and reading processes was taken into simulation to analyze the performance”. Unclear sentence.

Demonstration of near-sensor computing: handwriting recognition. Noise-aware training. Adding noise to the training data before regression is equivalent to Ridge Regression. It would seem to me easier to use Ridge regression with an appropriate parameter. If this is indeed equivalent to Ridge Regression, then please don't introduce a new terminology for something already existing.

Methods. Power estimation. I find this section unclear.

-Eq. 11 does not have any dependence on the number N of neurons, only on M .

-“The M parallel eRNRs can share one counter but the power for the other components increases.”

Increases with what ? Unclear

-What would be the best speed at which to operate the system? 50ns per operation to be compatible with the memristor array?

Fig 4.

-Panels c and d. Please add in caption that this is 1 step ahead prediction.

-Panel e and f. Unclear how these are computed. Please clarify.

Reviewer #3 (Remarks to the Author):

In the manuscript “Rotating neurons for all-analog implementation of cyclic reservoir computing” by X. Liang, Y. Zhong, J. Tang, et. al., the authors adopted a cyclic reservoir computing architecture which they show can be efficiently implemented using rotating neurons reservoir integrated with analog memristor array. They prove the equivalence between software simulation and hardware implementation. A proof-of-concept prototype of RNR was developed and to demonstrate near-sensor computing. The novel hardware design is tested on benchmarks, observing excellent performance in tasks such as nonlinear and Mackey-Glass chaotic time series prediction as well as in handwriting recognition.

Overall, the manuscript describes a novel physical design principle and hardware implementation that adopts a cyclic reservoir computing architecture. The works in this manuscript are systematic and well organized, representing an interesting and exciting progress towards practical RC systems for real-time signal processing in applications. I recommend publication in Nature Communications, subject to some of the technical questions be addressed properly - as follows.

1. The proposed physical implementation relies on a simple (yet effective) cyclic reservoir structure. The underlying hypothesis seems to be that random RCs are not as easily realizable physically. In between pure random and cyclic RC, and from a more fundamental and basic perspective, can the authors discuss and comment on what would be the class of RC structures that can similarly be mapped to efficient physical designs?
2. Suitable mask matrix can enhance the nonlinear dynamics in the reservoir. What is the influence of mask matrix (or the input layer matrix) on the proposed RNR? Will the performance be enhanced if chosen a non-binary mask matrix?
3. Could the authors discuss more about the impact of the consistency between each nonlinear node on the system performance?

** See Nature Research’s author and referees’ website at www.nature.com/authors for information about policies, services and author benefits.

Response Letter to reviewers' Comments

The authors sincerely appreciate the reviewers' insightful and constructive comments. We have carried out additional experiments and revised the manuscript according to the reviewers' suggestions. Below are the point-by-point responses to each comment. All the changes to the manuscript are marked in red.

Reviewer #1

General comments:

This manuscript reports on a study demonstrating a novel type of reservoir computing based on a cyclic reservoir. The study is noteworthy in two aspects:

- (i) it is the first to demonstrate a hardware realization of the concept of a cyclic reservoir, showing its equivalence with a rotating neuron reservoir device;*
- (ii) the novel hardware prototype developed in this study demonstrates a low-power end-to-end analog system capable of online learning.*

The work is highly original and has potential to significantly impact the field of neuromorphic computing. It is refreshing to see such a compelling demonstration of physical reservoir computing. In my opinion, the novelty of the work and quality of the results merit publication in NCOMMS. The supplementary videos are impressive. This notwithstanding, I do have some suggestions below for improving the manuscript.

Response:

We are grateful for the reviewer's positive comments and recognizing the novelty and importance of our work in the field of neuromorphic computing. We have further revised the manuscript following the reviewer's suggestions as detailed in the following.

Comment #1:

Discussion: The discussion as currently presented is a summary of the results. I would like to see this rewritten so that the authors discuss their results, both qualitatively and quantitatively, in the context of other similar studies. e.g. how do their NARMA10 simulation results compare to those reported by Appeltant et al. (2011) for a single cyclic reservoir? What are the similarities/differences? Similarly, how do their experimental results for the Mackey-Glass forecasting task compare to other physical RC studies? Can the authors comment on the ability to predict more chaotic MG signals (e.g. $\tau \gg 17$)? So, instead of predicting more steps ahead, predict one step ahead, but for a more chaotic signal. See also this paper that shows how training on more chaotic MG signals can improve prediction of less chaotic signals (i.e. demonstration

Response:

We thank the reviewer for raising this important point. Following your suggestions, we have added more qualitative discussions and comparisons in the revised **Discussion** section. Compared to literature reports, our simulation result on NARMA10 achieves the best performance in terms of NRMSE (0.078 for single eRNR and 0.055 for parallel eRNRs), to the best of our knowledge. Appeltant et al. (2011) claimed that their result (NRMSE=0.15) was the best one using hardware-based model at that time. Meanwhile, it has also been found that an earlier work¹ using a software-based echo state network achieved NMSE = 0.0098 (equivalent to NRMSE of 0.099). It can be seen that our NRMSE values are clearly lower than both prior works. Such improvement in our work can be mainly attributed to the extra nonlinearity provided by the diode model in our hardware-based simulation, which is different from the abrupt ON/OFF in the software-based ReLU function. In addition, the Mackey-Glass (MG) forecasting task preliminarily tested the computing ability of our eRNR prototype under different experimental setups and numbers of parallel eRNRs. The main advantages of our prototype compared with other physical implementations are the system simplicity and the novel architecture. As the reviewer suggested, we have now tested the prototype's performance with more chaotic MG signals ($\tau \gg 17$), and the results show that the system still works properly for one-step ahead prediction of more chaotic MG signals (**Supplementary Figs. 1a-f**). Meanwhile, for sanity check, we have also evaluated the system performance for 20-steps ahead prediction, which degrades more obviously as τ increases (**Supplementary Fig. 1g**). This result is also consistent with the paper by Zhu et al. (doi: 10.1109/ICRC2020.2020.00007) suggested by the reviewer.

Changes in the manuscript:

Page 14, Line 352: It is found that the additional nonlinearity provided by the hardware-based dynamic neuron could enhance our system performance on approximating NARMA10 system, which demonstrates the computing potentials of the proposed RNR method.

Page 15, Line 358: This experiment result further validates the computing ability of our eRNR prototype under different experimental setups.

Page 10, Line 242: Moreover, our experiment also revealed the eRNR prototype can properly predict more chaotic signals ($\tau \gg 17$) one step ahead (**Supplementary Fig. 1a-f**). In comparison, the system performance could degrade as τ increases when predicting more steps ahead (**Supplementary Fig. 1g**).

Added reference:

32 Zhu, R. *et al.* Harnessing adaptive dynamics in neuro-memristive nanowire networks for transfer learning. In *2020 International Conference on Rebooting*

Supplementary Figure 1:

Supplementary Figure 1. Experimental results for Mackey-Glass chaotic signal prediction with $\tau > 17$. a, b, c, Three episodes of one-step ahead prediction of Mackey-Glass time series result compared with the ground truth using the chaotic signal with $\tau =$ (a) 20, (b) 35 and (c) 50. d, e, f, Phase space of the prediction compared with ground truth using the chaotic signal with $\tau =$ (d) 20, (e) 35 and (f) 50. g, NRMSE results of 1 and 20 steps ahead prediction with varied τ values.

Comment #2:

On line 347-348: please cross-reference supp table 1. Also, please add clarification that the stated power (32.7 μ W) is for 10 Hz while the memristor array dominates at higher processing rates. Can the authors similarly compare to other memristor-based RC systems?

Response:

Thank you for your comment. We have now cross-referenced **Supplementary Table 1** in the revised **Discussion** section. Our power analysis shows that the static power, mainly dissipated by the dynamic neurons, dominates the system for processing rates lower than 100 kHz, while the overall system power remains at a low level for higher processing rates. Here we have added **Supplementary Table 2** to show the power breakdown at different processing frequencies according to **Eq. (11)**. This result can be explained by the fact that most computations occur in the analog domain that only contribute to static power, which is in line with the advantage of analog neuromorphic computing.

$$P = P_c + \left(P_s + P_t + \frac{E_c^{dyn} + E_t^{dyn}}{\tau_r} \right) \times M + \frac{E_m^{dyn}}{\tau_r} \quad (11)$$

In addition, we have also tried to compare the power consumption of our eRNR system with memristor-based reservoir computing systems reported in literature^{2,3}. However, their **system-level power consumptions** were not reported in most studies, which mainly focused on the device-level power consumption instead. So we listed all the previous studies in **Supplementary Table 1** that have clearly reported how much power is needed to run their reservoir computing system.

Supplementary Table 1. Comparison with literature-reported reservoir systems

Reference	Implementation	N	Processing rate (Hz)	Power
Alomar et. al. ⁴	FPGA	48	10^6	1.5W
Kleyko et. al. ⁵	FPGA	100	-	1.6W
Alomar et. al. ⁶	FPGA	50	1142	83mW
Brunner et. al. ⁷	Optoelectronic	388	13×10^6	150W
This work	All-analog eRNR	64	10	32.7 μ W
			1×10^3	32.9 μ W
			100×10^3	79.0 μ W
			10×10^6	4.7mW

Changes in the manuscript:

Page 15, Line 363: The overall system power consumption was estimated to be as low as 32.7 mW **for the handwriting tasks operating at 10Hz ($\tau_r = 0.1s$)**, which showed an advantage of more than three orders of magnitudes compared to literature-reported reservoir computing systems. **Also, further power analysis suggests that the static power, mainly dissipated by the dynamic neurons, dominates the system for processing rates lower than 100 kHz, while the overall system power remains at a low level for**

higher processing rates (>100kHz) (see **Supplementary Table 1**). This result can be explained by the fact that most computations occur in analog domain that only contributes to the static power at low frequency, which is in line with the advantage of analog neuromorphic computing. The dynamic power, mainly attributed to logic switches and memristor array, starts to dominate for processing rates higher than 100kHz (see **Supplementary Table 2**). Further discussions on the low-power advantage of eRNR can be found in **Supplementary Note 3**.

Page 23, Line 530: The power breakdown at different processing frequencies is shown in **Supplementary Table 2**.

Supplementary Table 2. Power breakdown for 8 × 8 eRNR system (μW)

Processing rate (Hz)	eRNR			Memristor	Total power (μW)
	Counter	Rotor	Neurons		
10	0.93	5.59	26.16	46.3×10^{-4}	32.7
1×10^3	0.93	5.59		46.3×10^{-2}	32.9
100×10^3	0.96	5.64		46.3	79.0
10×10^6	3.98	11.03		4633.6	4674.8

Comment #3:

Lastly, the authors should mention prospects for future studies that expand the capabilities of their system. e.g. online classification of more than 5 classes.

Response:

Thank you for pointing this out. There are several techniques to improve the RNR system capabilities to deal with more complex tasks (such as classification of more than 5 classes): 1) increase the number of neurons (N) to enhance both MC and network size; 2) increase the number of parallel RNR (M) to enhance the network size; 3) using different configurations for each neuron (currently they are simply replicated) is expected to enhance the state richness, and they could be optimized for different tasks. For example, higher τ_n could enhance the system performance on more chaotic signals; 4) develop a deep eRNR, consisting of multiple eRNR cells in series, could enhance the classification ability for different classes input. We have incorporated these performance enhancement techniques for future studies in the **Discussion** section to provide prospects for future studies.

Changes in the manuscript:

Page 15, Line 375: To further enhance the eRNR system capabilities to deal with more complex tasks, a useful approach would be increasing the number of neurons (N) or the number of parallel eRNRs (M) to expand the network size. Furthermore, a deep eRNR,

consisting of multiple eRNR cells in series, could enhance the classification performance for input of different classes. Also, from hardware perspective, different configurations of each neuron could be beneficial to enhance the state richness and hence improve the system performance. In addition, the eRNR design can be miniaturized and monolithically integrated on chip for low power and ultrafast computing. It is also worth mentioning that the dynamic neuron may be replaced by recently reported emerging devices (e.g., dynamic memristors^{2,8}, spintronics⁹) to further reduce the size and power consumption.

Minor Comment #1:

line 44: the LSM-based RC work of Wolfgang Maass should also be acknowledged (Maass et al. 2002).

Response:

We thank the reviewer for this good point. Indeed, the pioneer work of Wolfgang Maass discussed LSM RC should be acknowledged.

Changes in the manuscript:

Reference added:

- 3 Maass, W., Natschläger, T. & Markram, H. Real-Time Computing Without Stable States: A New Framework for Neural Computation Based on Perturbations. *Neural Comput.* **14**, 2531-2560, (2002).

Minor Comment #2:

line 61: please cite RC studies using memristive nanowire networks (e.g. Lilak et al. 2021 <https://doi.org/10.3389/fnano.2021.675792> and refs therein).

Response:

We thank the reviewer for this comment. The reservoir computer based on memristive nanowire is also a novel and advanced implementation that should be cited.

Changes in the manuscript:

Reference added:

- 19 Lilak, S. *et al.* Spoken Digit Classification by In-Materio Reservoir Computing With Neuromorphic Atomic Switch Networks. *Frontiers in Nanotechnology* **3**, 38, (2021).

Minor Comment #3:

lines 117, 119 (and possibly elsewhere): will be  is

Response:

We have corrected the gramatic errors as the reviewer suggested. Thank you.

Minor Comment #4:

line 128: please cross-reference Methods when mentioning "fundamentals of RNR".

Response:

We have cross-referenced to the **Methods** section as the reviewer suggested. Thank you.

Minor Comment #5:

lines 141-143: needs rephrasing, e.g. The simulator was developed to evaluate the performance of the eRNR and demonstrate its equivalence to a CR (as shown analytically in Methods).

Response:

We have rephrased the sentence as the reviewer suggested. Thank you.

Changes in the manuscript:

Page 6, Line 141: Moreover, a simulator was developed to evaluate the performance of the eRNR under different configurations and demonstrate its equivalence to a CR (as proved analytically in **Methods**). The first simulation was designed to confirm the consistency between RNR and CR and emphasize the role of rotation in RNR. The key network characteristics under different parameters, nonlinearity and rotation directions were investigated.

Minor Comment #6:

line 171: NARMA  NARMA10 (because lower orders aren't chaotic - in fact, I'm not entirely sure even NARMA10 is technically chaotic, although it's certainly highly nonlinear)

Response:

Thank you very much for the comment. As the reviewer correctly pointed out, NARMA is not a chaotic system since the output is insensitive to the initial condition. Only a certain number of previous inputs are dependent as defined by the equation. We have removed the 'chaotic' from the sentence.

Changes in the manuscript:

Page 7, Line 170: the eRNR should be able to approximate a nonlinear chaotic system.

Minor Comment #7:

line 177: typo with NARMA10

Response:

Thank you very much for pointed out the typo. We have corrected it in the revision.

Minor Comment #8:

line 184: can be  is

Response:

We have corrected it as the reviewer suggested. Thank you.

Minor Comment #9:

lines 192 -194: this strong statement needs to be backed up with references and discussed in the Discussion (as per my comment above).

Response:

We thank the reviewer for raising this good point. We have carefully checked related literature. For the NARMA10 system, the article¹⁰ that proposed delay-based reservoir computing mentioned that their result (NRMSE=0.15) was the best using hardware-based model. Meanwhile, it has also been found that an earlier article¹ using software echo state network can reach NMSE = 0.0098 (equivalent to NRMSE of 0.099). Here we have now cited these two papers to support our statement in the revision.

Changes in the manuscript:

Reference cited at Page 8, Line 196: To the best of our knowledge, the NRMSE values for both single eRNR (0.078) and parallel eRNR (0.055) are the lowest compared with the previous studies^{1,10} in the field of RC.

Minor Comment #10:

line 194: non-ideality  nonlinearity

Response:

Thank you for your comment. We agree that “nonlinearity” is a better word to describe the reason for the record-low NRMSE value achieved in this work.

Changes in the manuscript:

Page 8, Line 198: The reason is that the **exponential type of nonlinearity provided by the transition region of diode (differ from the ideal ON/OFF in the software ReLU function) enhances the state representation of NARMA10 system.**

Minor Comment #11:

line 207: remove "it has been studied that"

Response:

We have removed “it has been studied that” as the reviewer suggested. Thank you.

Minor Comment #12:

lines 238-239: cite references to back up this statement

Response:

We thank the reviewer for this comment. We have now cited three papers using spintronic and memristive devices (Torrejon, J. et. al., Nature, 2017; Moon, J. et. al., Nature Electronics, 2019; Zhong, Y., et. al., Nature Communication, 2021) to back up our statement on the low power consumption.

Changes in the manuscript:

Reference cited at Page 10, Line 247: In literature, the previously reported reservoir computing demonstrations can reach rather low power consumption for certain parts inside the system using novel devices and materials^{2,8,9}.

Minor Comment #13:

line 361: remove "or backpropagation"

Response:

Thank you. We have removed the word ‘backpropagation’ as the reviewer suggested.

Minor Comment #14:

line 378: remove "which is of great interest"

Response:

We have corrected it as the reviewer suggested. Thank you.

Minor Comment #15:

lines 479-480: remove "the static power" and remove repeat of $P_s=3.27\mu W$.

Response:

We have corrected them as the reviewer suggested. Thank you.

Minor Comment #16:

line 593: Fig. 1 caption, possibly try placing the legend at the bottom because it applies to each fig

Response:

We thank the reviewer for this good suggestion. We have corrected the figure caption.

Changes in the manuscript:

Fig. 1:

Page 30, line 671, end of Figure 1 caption: **The legend for all subfigures is provided at the bottom.**

Minor Comment #17:

line 616: Fig. 2b caption, mention $a(k)$ is neuron output at k -th step

Response:

We have mentioned that $a(k)$ is neuron output at k -th step as the reviewer suggested. Thank you.

Changes in the manuscript:

Page 31, Line 686: **b**, A general schematic for the dynamic properties required to be the neuron in an RNR. On the arrival of a neuron input $\mathbf{R}^k \mathbf{W}_{in}(k)$ that has been processed by pre-neuron rotor and input weights, the neuron performs nonlinear transform f , integration (feedback line), and leakage (decay factor d) operations on the signal. **The $a(k)$ is neuron output at k^{th} step.**

Minor Comment #18:

line 626: Fig. 3 caption, mention these are simulation results

Response:

We have clarified it as the reviewer suggested. Thank you.

Changes in the manuscript:

Page 33, Line 696: Figure 3. **Simulation results on** network characteristics of eRNR and its performance in time series prediction.

Minor Comment #19:

line 646: Fig. 4 caption, clarify to $M=8$ and $N=8$

Response:

We have clarified it as the reviewer suggested. Thank you.

Changes in the manuscript:

Page 35, Line 717: **Figure 4. 8×8 eRNR prototype for Mackey-Glass time series prediction.** **a**, An eRNR prototype consists of eight 8-neuron eRNRs (*i.e.*, $M = 8$ and $N = 8$).

Reviewer #2

General comments:

Report on Rotating neurons for all-analog implementation 1 of cyclic reservoir computing

In this work, the authors realize an analog electronic implementation of cyclic reservoirs.

On the plus side:

-cyclic reservoirs have to my knowledge not been implemented before (while reservoirs with delay lines, which are very similar, have been extensively studied).

-the method to implement cyclic reservoirs, using a rotating logic element, is novel and elegant.

-the system is fully analog, including an analog output layer. An interface with a tactile screen is presented.

-An estimate of the energy consumption of an integrated system using CMOS technology is presented.

-Performance on tasks is good.

-The paper is mostly well written and easy to follow.

Response:

We are grateful to the reviewer for the thoughtful comments and recognizing the merits of this works. In this following, we have addressed your detailed comments point by point and revised the manuscript accordingly. Hopefully the revised version and our responses can fully address the reviewer's concerns.

Comment #1:

The trick of using a rotating logic element is a trick, not an important conceptual advance.

Response:

We thank the reviewer for this comment. The community of neuromorphic computing has been actively pursuing resource-efficient hardware to implement brain-like learning algorithms. The major conceptual advance of our work is that we **have proposed a novel rotation-based hardware architecture to implement reservoir computing**. We have also proved **excellent network-level consistency (see Fundamental of RNR, Methods) between the software algorithm (CR) and physically rotating object**, and therefore we can empower rotating objects with computing capabilities. The eRNR system composed by logic elements is just a prototype example to demonstrate the concept of RNR, which can be readily extended to other rotating objects. In our opinion, the conceptual advance of RNR could provide an advantageous paradigm to **explore**

the computing resource from physics. Meanwhile, it is also a practical solution for engineering a resource-efficient processor at the edge, which could find appealing applications in near-sensor or in-sensor computing where the all-analog processor can act as a direct interface of a sensory signal without any intermediary modules.

With the above conceptual advances, we believe our work could make a valuable contribution to the field of neuromorphic computing.

Changes in the manuscript:

Page 14, Line 342: In summary, we have developed a novel RNR architecture for all-analog neuromorphic computing for the first time, **which represents a fundamentally different reservoir architecture compared to conventional hardware implementations.** The proposed RNR has been validated in theory, simulation, and experiment. The theoretical analysis of RNR rigorously mapped the CR algorithm onto the physical rotation of dynamic neuron array, laying a solid foundation for the following hardware implementation. **Such RNR can be embedded into natural rotating components in various electronics, mechanical systems or even nanorobotics and empower them with computing ability.**

Comment #2:

The cyclic reservoir is only a minor variation on delay reservoirs, which have been much studied.

Response:

We thank the reviewer for this comment. In our opinion, the cyclic reservoir, which is equivalent to the proposed rotating neuron reservoir (RNR) in this work, is fundamentally different from delay-based reservoirs. In fact, delay-based reservoir computing proposed in 2011¹⁰ was also inspired by the cyclic reservoir¹¹ (see the Supplementary Information of (Appeltant et. al., Nature Communication, 2011)¹⁰) and has since gained lots of attention because it is friendly to hardware implementations. The relation between classical random RC, cyclic RC, delay-based RC and RNR are illustrated in **Fig. R1**.

Software RC algorithm

Hardware RC architecture

Figure R1. Relation between classical random RC, cyclic RC, delay-based RC and RNR.

As we can see from the comparison in **Table R1**, delay-based RC should be considered in parallel to RNR since they represent completely different implementation paradigms, while both are inspired by cyclic reservoir which is a simplified version of classical reservoir.

Table R1. Comparison between delay-based reservoir and RNR

Differences	Delay-based reservoir	RNR
Signal flow	In series	In parallel
Core	Single neuron + virtual nodes	Multiple neurons
Preprocessing	Time-multiplexing	Not required
Memory	Delay line + feedback	Rotator (logic elements)

Although delay-based reservoir computing has been well studied, its weakness hinders its further development as discussed in the **Introduction** part (Page 3, Lines 57-69). The highly cited review article on physical reservoir computing¹² by Tanaka et al. (2019) also pointed out that implementing a delayed feedback loop is not a straightforward task. A novel resource-efficient implementation is still of great interest in the field of reservoir computing and neuromorphic computing, which is what we have demonstrated in this work with RNR.

Common #3

A convincing argument for using cyclic reservoirs with rotating elements, rather than delay reservoirs, is not presented.

Response:

Thanks for the comment. In the **Introduction** part, we have already discussed the major drawbacks associated with the use of delayed feedback in delay-based reservoirs (Page 3, Lines 57-69), which motivates us to propose a novel resource-efficient implementation, the cyclic reservoir (CR) with rotating elements. In the revision, we now make more comparisons between the proposed RNR with existing reservoir implementations including delay-based one (see Line 75-77, Line 343-344, Line 330-332, Line 365-366, Supplementary Note 3). The main advantages of using RNR can be summarized as follows:

- **Explainability.** In the **Methods** section, we have proven the equivalence between software CR and rotation-based framework, which is also endorsed by the task-independent properties analysis in Fig. 3. Such explainability, which is currently lacking for delay-based reservoir, would benefit future developments and optimizations. **For delay-based reservoir**, a nonlinear node is used to provide nonlinear function while its dynamic property makes connections between neighboring virtual nodes. Additionally, one or more external delayed feedback lines merge current input and previous state matrix to provide memory capacity. Appeltant et al. (2011) also mentioned in their supplementary information that the working principle is different from the traditional reservoir computing. Therefore, delay-based dynamical system can act like a reservoir, but it is less likely to be explained by a standard software reservoir computing model.
- **Hardware implementation.** The advantage of RNR in hardware implementation has been discussed in Line 366, Line 424 (added), and Supplementary Note 3 (added). In our method, a physical reservoir computer can be implemented by rotating elements. The electrical RNR can be achieved by low-cost logic elements without other critical components, such as analog-to-digital converter (ADC) and memory units. **For delay-based reservoirs**, the delayed feedback line plays a crucial role in generating memory capacity. However, the electrical implementation of delayed feedback usually requires ADC, digital-to-analogue converter (DAC) and memory, as presented in¹⁰. These components are usually not desirable in neuromorphic computing since they (1) require additional space and cost; (2) consume more power, especially in high processing speed or large network size; (3) require additional control unit for reading, erasing and writing the memory as well as ADC/DAC, which increases the system complexity. More comprehensive discussions about these digital memory-related constraints in neuromorphic computing can be found in the literature^{9,13,14}.
- **Parallel computing.** RNR is a highly parallel architecture (see Fig. 2 and Supplementary Videos 1 and 2). In comparison, **for delay-based reservoir**, the use

of time-multiplexing leads to serial operation at both input and readout; therefore, only one state value can be obtained at each time step. The benefits of parallel computing can be found in the literature¹²⁻¹⁴.

- **Power consumption.** Because of the simplicity of eRNR, it exhibits excellent power efficiency. According to the implementation of eRNR (Fig. 2), all the components that consume power are discussed (see **Methods**). The comparison with delay-based reservoir and other implementations is listed in **Supplementary Table 1**. To the best of our knowledge, our implementation shows the record-low **power consumption for the entire system**, which highlights the power advantage when using eRNR.

Of course, we have to emphasize that the delay-based reservoir and cyclic reservoir are not necessarily exclusive, but instead they can be supplementary for each other. For example, one can add a delayed feedback line to RNR to further enhance the memory capacity. Such a hybrid rotation and delay reservoir is feasible and warrants future exploration.

Here we summarize the above comparison in **Table R2**:

Table R2. Comparison between delay-based reservoir and RNR

	Delay-based reservoir	RNR
Explainability	Not explainable by standard RC	Equivalent to cyclic reservoir
Hardware Implementation	Time multiplexing + Dynamic neuron + Delayed feedback line (A/D conversion involved)	Dynamic neurons + Rotators (low power logic elements)
Signal flow	In series	Parallel computing
Power consumption	Limited by the delay unit (~mW scale power consumption by the delay unit only)	Low power for the entire system (in the μ W scale)

Changes in the manuscript:

Page 3, Line 75: Compared with the existing implementations of reservoir computing^{2,15-18}, RNR is more hardware-friendly, resource-efficient, fully parallel and explainable by standard CR.

Page 14, Line 343: In summary, we have developed a novel RNR architecture for all-analog neuromorphic computing for the first time, which represents a fundamentally different reservoir architecture compared to conventional hardware implementations.

Page 18, Line 424: It implies that a rotating object with dynamic neurons can act as a reservoir computer without using extra control units, ADC and memory, which

remarkably reduce the system complexity and power consumption compared with conventional hardware implementation (see **Supplementary Note 3**).

Comment #4:

The analog output layer has been presented in previous work.

Response:

Thanks for the comment. As the reviewer pointed out, memristor crossbar has been presented in previous works to implement the analog fully connected output layer for reservoir computing. In fact, memristor has been widely used to implement fully connect layers in artificial neural network by taking its advantage of computing-in-memory capability¹⁹⁻²². Similarly, memristor arrays have also been used in recently reported reservoir computing demonstrations to implement the final output layer^{23,24}. However, we shall clarify that this is not the main focus and key novelty of our work, which instead aims to demonstrate a novel **rotation-based reservoir computing**. The memristor crossbar-based analog output layer used in our eRNR system is rather for the purpose of demonstrating all-analog neuromorphic computing. To clarify this point, we have added a statement in Line 499.

Changes in the manuscript:

Page 21, Line 499: Memristor-based analog computing shows great potentials in neuromorphic computing. **While the input and reservoir layer have been achieved by eRNR design, the output layer, which employs standard vector-matrix multiplication operations, can be effectively implemented by a memristor array for end-to-end all-analog computing.** The description of the memristor array can be found in **Supplementary Fig. 2**.

Reference added:

- 42 Yu, J. *et al.* Energy efficient and robust reservoir computing system using ultrathin (3.5 nm) ferroelectric tunneling junctions for temporal data learning. In *2021 Symposium on VLSI Technology*. 1-2 (IEEE, 2021).
- 43 Milano, G. *et al.* In materia reservoir computing with a fully memristive architecture based on self-organizing nanowire networks. *Nat. Mater.*, published online (2021).

Technical Comment #1:

The prototype system is not presented in detail. Please provide enough details for another researcher to be able to reproduce the experiment.

Response:

We thank the reviewer for this thoughtful comment. The details about the prototype system have been now added to **Supplementary Note 2**.

Changes in the manuscript:

Supplementary Note 2, Line 110: The schematic of eRNR is shown in Fig. 2. The network size of our prototype is $N = 8$ and $M = 8$, which means the single eRNR consists of 8 neurons and there are 8 parallel eRNRs. Both pre- and post-neuron rotors were implemented by eight CD4051B which is an 8-channel analog multiplexer from Texas Instrument. The three signal selection ports were connected to a 3-bit binary counter consisted of a 4-bit counter (74LS161) and an inverter (74HC04). The input mask was implemented by 8 switches to select positive or negative signals. In order to improve the state richness, each eRNR circuit should use a different input mask configuration.

Technical Comment #2:

The paper is not very clear (particularly in the abstract and introduction) on what is experimental, what is simulation, and what is theoretical calculation.

Response:

Thank you very much for the comment. We have now clarified how the results were obtained in the revised manuscript: (1) the equivalence between software and hardware was proven mathematically; (2) the record-low performance on NARMA10 was obtained in the simulation using eRNR model; (3) Mackey-Glass chaotic signal was tested in hardware experiment; (4) handwritten vowel recognition was demonstrated in hardware experiment; (5) power consumption was estimated using 65nm CMOS design in Cadence.

Changes in the manuscript:

Page 2, Line 26: The equivalence between the rotating neuron reservoir and standard cyclic reservoir algorithm is mathematically proved. Simulation shows the rotating neuron reservoir achieved record-low errors in the time-series prediction benchmark.

Page 4, Line 79: Furthermore, a prototype of eRNR composed of eight parallel reservoir circuits was built to demonstrate analog near-sensor computing, where Mackey-Glass time series prediction and real-time handwriting recognition were successfully performed **in hardware experiment**.

Page 4, Line 86: Finally, the CMOS circuit simulation based on standard 65nm technology indicated that the eRNR system consumed as low as $32.7\mu\text{W}$ for the handwriting recognition task, which was more than three orders of magnitudes lower than literature-reported reservoir systems.

Page 33, Line 697: Figure 3. **Simulation results** on network characteristics of eRNR and its performance in time series prediction.

Technical Comment #3:

If the authors claim that, in the context of CMOS implementations, their rotating architecture is more efficient than other architectures. They should present evidence. For instance it is not clear why a delay system with a single nonlinear node (Ref 6 Appeltant et al) could not be as energy efficient.

Response:

We thank the reviewer for this comment. The reasons for the high efficiency of RNR implementation can be explained as follows:

1. In the proposed rotating neuron architecture, the CR algorithm highly matches the hardware behavior. This excellent consistency frees the system from using an extra control unit, ADC and memory, which significantly reduces the system complexity and power consumption. Meanwhile, the rotation implemented by logic elements consumes extremely low power (in pW scale). In the delay-based reservoir presented by Appeltant et al.¹⁰, a continuous analog signal needs to be delayed for a certain time length. In their experiment, the delay line was implemented by a PC-controlled NI-6025E with 12-bit AD/DA from National Instrument, which alone consumes power >5W according to the datasheet, much higher than that of the logic elements for rotation. For a more practical comparison, we can roughly estimate the power needed to implement a typical delay line using CMOS components: 8-bit ADC (ADC1175-50), 8-bit DAC (DAC084S085) and 8-bit static random-access memory (SRAM). As can be found from their datasheet, their static power are all in milliwatts scale (i.e. ADC is 5mW, SRAM is ~6.6mW and DAC is ~1.1mW). In practical implementations, this delay line should be subjected to a controller or logic circuit to control the timing and digital address. Also, extra modulation units are essential in both the input and output layer for the serial time-multiplexing operation and taking the summation between the input signal and feedback signal. These components would also consume considerable power. In addition to the ADC-SRAM-DAC solution, there exists an analog CMOS-based solution for delaying a signal named bucket-brigade delay line invented by Philips Research Labs^{25,26}, which has been discontinued in last century. Its datasheet (MN3004) suggests that it consumes 165mW power. These values indicate that, even if we only consider the static power of **delay line**, the delay-based approach is significantly more power-hungry than eRNR.
2. From a more fundamental perspective, the **different mechanisms of introducing memory** in delay-based and rotation-based methods determine their potential in

power efficiency. In the delay-based approach, the memory is actually separated from the processor. Although the processing is carried out in a nonlinear dynamic node, the memory is mainly provided by the delay unit which is constrained by the limitations of conventional digital computing, such as power consumption, throughput and latency¹³. In the rotation-based method, the memory is provided by the rotating dynamic node itself (see **Fig. 3a** and **Methods**). Implementing the logic switches for rotation is much more resource-efficient than delay lines in terms of power and cost. Meanwhile, the rotating dynamic node serves to process the signal and retain previous information simultaneously. Such in-memory computing paradigm is advantageous for low-power computing. These fundamental differences result in the higher power efficiency of the rotation-based implementation in this work.

3. In addition, as we responded to Comment #3 earlier, the comparisons with delay-based reservoir and other implementations are listed in **Supplementary Table 1**. To the best of our knowledge, our implementation shows the record-low **power consumption for the entire system**, which highlights the power advantage when using eRNR.

Supplementary Table 1. Comparison with literature-reported reservoir systems

Reference	Implementation	N	Processing rate (Hz)	Power
Alomar et. al. ⁴	FPGA	48	10^6	1.5W
Kleyko et. al. ⁵	FPGA	100	-	1.6W
Alomar et. al. ⁶	FPGA	50	1142	83mW
Brunner et. al. ⁷	Optoelectronic	388	13×10^6	150W
This work	All-analog eRNR	64	10	$32.7\mu\text{W}$
			1×10^3	$32.9\mu\text{W}$
			100×10^3	$79.0\mu\text{W}$
			10×10^6	4.7mW

Changes in the manuscript:

Page 14, Line 338: **More discussions and comparison on the power efficiency of eRNR can be found in Supplementary Note 3.**

Page 15, Line 366: **Also, further power analysis suggests that the static power, mainly dissipated by the dynamic neurons, dominates the system for processing rates below 100 kHz, while the overall system power remains at a low level for higher processing rates (>100kHz) (see Supplementary Table 1). This result can be explained by the fact that most computations occur in the analog domain that only contributes to static power, which is in line with the advantage of analog neuromorphic computing. The dynamic power, mainly attributed to logic switches and memristor array, start to dominate the**

system for processing rates higher than 100kHz (see **Supplementary Table 2**). Further discussion on the low-power advantage of eRNR can be found in **Supplementary Note 3**.

Page 18, Line 424: It implies that a rotating object with dynamic neurons can act as a reservoir computer without using extra control units, ADC and memory, which remarkably reduce the system complexity and power consumption compared with conventional hardware implementation (see **Supplementary Note 3**).

Supplementary Note 3. Why eRNR can be more power efficient?

From a fundamental perspective, the different mechanisms of introducing memory in rotation-based architecture and other architectures largely determine their power efficiency. In the rotation-based architecture, the memory is provided by the rotating dynamic node itself (see **Fig. 3a** and **Methods**). The excellent consistency between the rotation behavior and software algorithm frees the system from using extra control units, ADC and memory, which remarkably reduce the system complexity and power consumption. Also, implementing the logic switches for rotation is a resource-efficient approach by using CMOS-based transmission gates. Meanwhile, the rotating dynamic node serves to process signal and retain previous information simultaneously. Such in-memory computing paradigm is advantageous for low-power computing. In other architectures, such as the well-studied delay-based one, the memory is actually separated from the processor. Although the processing carried out in nonlinear dynamic node was a significant progress, the memory is mainly provided by the delay unit which is constrained by the limitations of conventional digital computing, such as power consumption, throughput and latency¹³. These fundamental differences result in the better power efficiency for the rotation-based architecture.

Compared with the classic random reservoir computing, the key difference of cyclic reservoir is the connection in the reservoir layer defined by \mathbf{W}_{res} . The \mathbf{W}_{res} of random reservoir is a randomly generated matrix with a proper spectral radius, while the cyclic counterpart is a shifted identity matrix which can be implemented in a more deterministic manner without performance degradation¹¹. In this work, it has been proven that the cyclic \mathbf{W}_{res} can be equivalent to a physical rotor (see **Methods**), while an effective physical counterpart of random \mathbf{W}_{res} is yet to be found, which remains an exciting challenge to be addressed for future studies.

Minor Comment #1:

I found it quite confusing to use the acronym RC for reservoir computer, and CR for cyclic reservoir computer. Initially while reading the paper, I thought CR was a typo.

Response:

Thanks for the comment. To avoid any confusion, we have changed all the acronym 'RC' back to 'reservoir computing' in the revision, while keeping the acronym 'CR' for 'cyclic reservoir'.

Changes in the manuscript:

All the acronym '*RC*' were changed into '*reservoir computing*'

Minor Comment #2:

“Through our noise-aware training method, the conductance variation of memristor array was accommodated and a high classification accuracy of 94.0% was achieved.” Imprecise. Classification of what?

Response:

Thanks for the comment. The accuracy of 94.0% was achieved for in the handwriting vowel recognition task, which has been clarified in the revision.

Changes in the manuscript:

Page 4, Line 83: Through our noise-aware training method, the conductance variation of memristor array was accommodated and a high classification accuracy of 94.0% was achieved **in the handwriting vowel recognition task.**

Minor Comment #3:

“Finally, the system benchmark indicated that the eRNR system consumed as low as 32.7 microW for the handwriting recognition task, which was more than three orders of magnitudes lower than literature-reported reservoir systems.” Isn't this a theoretical estimate. The sentence suggests it is an experimental result.

Response:

We thank the reviewer for this comment. Yes, the power consumption is a theoretical estimate using the circuit design with standard 65nm CMOS technology. We have clarified it in the text.

Changes in the manuscript:

Page 4, Line 86: Finally, **the CMOS circuit simulation based on standard 65nm technology** indicated that the eRNR system consumed as low as 32.7 μ W for the handwriting recognition task, which was more than three orders of magnitudes lower than literature-reported reservoir systems.

Minor Comment #4:

Performance benchmark of eRNR. « the eRNR should be able to approximate a nonlinear chaotic system, for which NARMA is a widely recognized benchmark task to test RC performance.” NARMA is a Nonlinear Auto Regressive Moving Average system,

not a nonlinear chaotic system.

Response:

Thanks for pointing it out. NARMA is indeed not a chaotic system since the output is insensitive to the initial condition. Only a certain number of previous inputs are dependent as defined by the equation. We have hence removed the word ‘chaotic’ from the sentence.

Changes in the manuscript:

Page 7, Line 170: the eRNR should be able to approximate a nonlinear **chaotic** system.

Minor Comment #5:

“It is worth mentioning that the biologically realistic time constant values (...) were used throughout our hardware implementation and simulation so as to interact with the environment in biological time scales.” Unclear. What is biological? Why is slow good? Is it the electronic neurons that should imitate biological neurons? Or the task (handwriting recognition) that must be on the time scale of humans?

Response:

We thank the reviewer for this comment. The reviewer is correct that the electronics are preferred to be on the time scale of the human for the specific task of handwriting recognition. In fact, the ‘biologically realistic time constant’ was originated from the literature¹³ (Indiveri, G. et al., Proc. IEEE, 2015). It has been discussed that, in neuromorphic processors, electronics that are directly interacting with the environment and natural signals could exhibit a much longer time constant (e.g., >millisecond scale) compared with a conventional digital processor (e.g. <nanosecond scale). Similarly, if we design the neurons with a short time constant (τ_n) in our handwriting application, the user will have to write fast to maintain the effect of memory capacity, which is not realistic. In general, the setting of the time constant in the implementation of reservoir computing systems depends on the specific tasks. To avoid confusion, we have further clarified this point in the revision.

Changes in the manuscript:

Page 8, Line 186: It is worth mentioning that, **in neuromorphic computing system, the electronics directly interacting with the environment and natural signals could exhibit a much longer time constant (e.g., >millisecond scale) compared with those of digital system¹³. A fast time constant could result in a shortage in MC to retain history information. Such** biologically realistic time constant values (τ_n and τ_r , from milliseconds to seconds scale) were used throughout our hardware implementation and simulation.

Page 22, Line 519: For our application interacting with real-time handwriting signal, the operation period τ_r is a relatively slower value (0.1s) to match the time scale of human.

Minor Comment #6:

“To the best of our knowledge, the NRMSE values for both single eRNR (0.078) and parallel eRNR (0.055) are the lowest compared with the previous studies in the field of RC.” Maybe add some references?

Response:

We thank the reviewer for raising this important point. We have carefully checked related literature. For the NARMA10 system, the article¹⁰ that proposed delay-based reservoir computing mentioned that their result (NRMSE=0.15) was the best using a hardware-based model. Meanwhile, it has also been found that an earlier article¹ using software echo state network can reach NMSE = 0.0098 (equivalent to NRMSE of 0.099). Here we have now cited these two papers to support our statement in the revision.

Changes in the manuscript:

Reference cited at Page 8, Line 197: To the best of our knowledge, the NRMSE values for both single eRNR (0.078) and parallel eRNR (0.055) are the lowest compared with the previous studies^{1,10} in the field of RC.

Minor Comment #7:

Eq. 2. The Mackey Glass equation has a power at the denominator. Eq. 2 seems wrong?

Response:

Thanks for pointing out the typo in Eq. 2. The power of n at the denominator was indeed missing. In addition, we suggest using the differential equation rather than the discrete approximation. We have corrected it in the revised manuscript.

Changes in the manuscript:

Page 9, Line 210:

The Mackey-Glass system is defined by:

$$\frac{dy}{dt} = \beta \frac{y(t - \tau)}{1 + y(t - \tau)^n} - \gamma y(t) \quad (2)$$

where the system parameters g , b , and n follow the widely used values 0.1, 0.2, and 10, respectively.

Minor Comment #8:

Demonstration of near-sensor computing: handwriting recognition .“Using the

datasets collected from the eight participants, the noise of our memristor array associated with the mapping and reading processes was taken into simulation to analyze the performance". Unclear sentence.

Response:

Thanks for pointing it out. Due to the conductance noise of memristor devices, the weight values cannot be accurately mapped onto the memristor, which has to be taken into consideration in the simulation in order to find a proper training scheme. We have completely revised the sentence to avoid confusion.

Changes in the manuscript:

Page 12, Line 300: **The next simulation evaluates the effect of conductance noise of memristors on the classification performance in order to find a proper training scheme.**

Minor Comment #9:

Demonstration of near-sensor computing: handwriting recognition. Noise-aware training. Adding noise to the training data before regression is equivalent to Ridge Regression. It would seem to me easier to use Ridge regression with an appropriate parameter. If this is indeed equivalent to Ridge Regression, then please don't introduce a new terminology for something already existing.

Response:

Thanks for your comment. As the reviewer mentioned, adding noise before regression could result in similar effect as Ridge regression on the weight matrix. In this work, by adding noise, we can systematically evaluate the effect of the noise amplitude as well as the distribution on the system performance, which enables us to match the memristor behavior in simulation and experiment, as shown in **Fig. 5h**. In fact, the idea and terminology of noise-aware training has been commonly used in literature for memristor-based neural networks²⁷⁻²⁹. Considering the main noise source in the all-analog architecture is the conductance noise of memristor devices, we suggest keeping the term of "noise-aware training" in this work and adding explanations to clarify it.

Changes in the manuscript:

Page 12, Line 303: **In our experiment, the intrinsic noise of memristors is the dominating noise source in the all-analog system. To achieve a high accuracy, we have adopted a noise-aware training method to obtain a more robust \mathbf{W}_{out} in the presence of memristor conductance variation^{27,29}.**

Reference added:

- 36 Joshi, V. *et al.* Accurate deep neural network inference using computational phase-change memory. *Nat. Commun.* **11**, 2473, (2020).
- 37 Kariyappa, S. *et al.* Noise-Resilient DNN: Tolerating Noise in PCM-Based AI

Accelerators via Noise-Aware Training. *IEEE Trans. Electron Devices* **68**, 4356-4362, (2021).

Minor Comment #10:

Methods. Power estimation. I find this section unclear.

-Eq. 11 does not have any dependence on the number N of neurons, only on M .

-“The M parallel eRNRs can share one counter but the power for the other components increases.” Increases with what ? Unclear

-What would be the best speed at which to operate the system? 50ns per operation to be compatible with the memristor array?

Response:

We thank the reviewer for raising these important points. For the first question, the power estimation aims to analyze the power consumed by M 8-neuron eRNR in the handwriting recognition and Mackey-Glass time series prediction task. Here the number of neurons is fixed at $N = 8$ in order to evaluate the computing ability dependence on M in the Mackey-Glass time series prediction. We have clarified it in the revision.

For the second question, the M parallel eRNR can share a common counter, so the static power consumed by the counter does not increase with M . This is the reason that the P_c is excluded in the multiplication with M in Eq. (11). Meanwhile, the dynamic power of counter and transmission gate, and static power of neuron and transmission gate are proportional to M . We have clarified it in the revision.

For the last question, the best speed to operate the system should actually depend on the specific application. As discussed above, the applications interacting with the natural environment in real-time usually have a slow processing rate. For example, our eRNR prototype was designed for handwriting recognition task. Both time constants for operating the system (τ_r) and neurons (τ_n) are chosen to be relatively slow (on the order of ~ 0.1 s). Meanwhile, it does not need to be compatible with the memristor array. During every τ_r , only one-time inference is needed since all state channels are monotonously increased or decreased. So the memristor array is activated once for ~ 50 ns during every τ_r to yield the output. More information about our memristor array can be found in our previous works^{19,21}. Furthermore, the operation speed can be much faster when the system acts as a data processing accelerator that servers to speed up computing and no need to interact with the natural environment, which remains future exploration. We have clarified this point in the revision.

Changes in the manuscript:

Page 22, Line 513: The total power consumption P of the system consisting of M 8-neuron eRNRs (where the number of neurons are fixed at 8) can be summarized as:

Page 22, Line 518: The M parallel eRNRs can share one counter but the power for other components increases with the number of parallel eRNRs M . For our application interacting with real-time handwriting signals, the operation period τ_r is a relatively slow (~ 0.1 s) to match the time scale of human.

Minor Comment #11:

Fig 4.

-Panels c and d. Please add in caption that this is 1 step ahead prediction.

-Panel e and f. Unclear how these are computed. Please clarify.

Response:

Thank your for the comment. We have added the one-step ahead prediction to the caption. In addition, the phase diagram was computed by plotting $y(t)$ for x -axis and $y(t-\tau)$ for y -axis, respectively. The $y(t)$ values are from the signal in Fig. 4c and d. The phase plot is a commonly used diagram to study chaotic signal and visualize the chaotic attractor.

Changes in the manuscript:

Page 35, Line 720: Two episodes of one-step ahead prediction of Mackey-Glass time series result compared with the ground truth using (c) one eRNR (NRMSE = 0.17) and (d) eight parallel eRNRs (NRMSE = 0.03).

Page 35, Line 724: The phase diagram was computed by plotting the prediction and ground truth series $y(t)$ for x -axis and $y(t-\tau)$ for y -axis, respectively.

Reviewer #3

Overall Comment:

In the manuscript “Rotating neurons for all-analog implementation of cyclic reservoir computing” by X. Liang, Y. Zhong, J. Tang, et. al., the authors adopted a cyclic reservoir computing architecture which they show can be efficiently implemented using rotating neurons reservoir integrated with analog memristor array. They prove the equivalence between software simulation and hardware implementation. A proof-of-concept prototype of RNR was developed and to demonstrate near-sensor computing. The novel hardware design is tested on benchmarks, observing excellent performance in tasks such as nonlinear and Mackey-Glass chaotic time series prediction as well as in handwriting recognition.

Overall, the manuscript describes a novel physical design principle and hardware implementation that adopts a cyclic reservoir computing architecture. The works in this manuscript are systematic and well organized, representing an interesting and exciting progress towards practical RC systems for real-time signal processing in applications. I recommend publication in Nature Communications, subject to some of the technical questions be addressed properly - as follows.

Response:

Thank you very much for your positive comments on our work. We have improved the manuscript as the reviewer suggested.

Comment #1:

The proposed physical implementation relies on a simple (yet effective) cyclic reservoir structure. The underlying hypothesis seems to be that random RCs are not as easily realizable physically. In between pure random and cyclic RC, and from a more fundamental and basic perspective, can the authors discuss and comment on what would be the class of RC structures that can similarly be mapped to efficient physical designs?

Response:

We thank the reviewer for raising this important question. In between pure random and cyclic RCs, and from a more fundamental and basic perspective, cyclic reservoir and RNR is an interesting pair that exhibits a high level of consistency (see **Methods**). Such high-level consistency means that a complete cyclic reservoir algorithm can be fully mapped onto the physical behavior of an RNR without extra components. In neuromorphic computing, such consistency is highly favored in order to map computing operations onto more efficient electronics. Recent examples include: 1)

using the computing-in-memory property of memristors to perform vector-matrix multiplication or multiply-accumulate operations^{21,22}; 2) using device's nonlinearity for nonlinear calculations¹⁰; 3) and also exploring device's oscillation⁹, phase change^{29,30} and stochastic response as computational resources. A neuromorphic system fully utilizing these physical properties can be more resource-efficient compared with digital implementations. However, these physical behaviors usually can only be **partially** mapped to the functionalities of a computing system. In most cases, extra components, such as ADC, DAC, memory and controller, are needed to run the entire system. Differently, the proposed RNR in this work maps the cyclic reservoir to a rotation-based object at the **architecture level**, which yields efficient physical design and unique advantages. Therefore, for a class of RC structure as well as other algorithms, it can be mapped to efficient physical design when its

Regarding the classic random RC (also known as echo state network), the key difference of cyclic reservoir is the connection in the reservoir layer defined by \mathbf{W}_{res} . The \mathbf{W}_{res} of random reservoir is a randomly generated matrix with a proper spectral radius, while the cyclic counterpart is a shifted identity matrix which can be implemented in a more deterministic manner without performance degradation¹¹. In this work, it has been proven that the cyclic \mathbf{W}_{res} can be equivalent to a physical rotor (see **Methods**), while an effective physical counterpart of random \mathbf{W}_{res} is yet to be found, which remains an exciting challenge to be addressed for future studies. Therefore, at the current stage, the random RC is less likely to be **similarly** mapped to efficient physical design as RNR. To clarify this point, we have provided detailed discussions to explain the fundamental reason for the power efficiency and why random RC is not easy to be similarly mapped to efficient design in **Supplementary Note 3**.

Table R3. Differences between the \mathbf{W}_{res} in random and cyclic reservoirs

	\mathbf{W}_{res} in random reservoir	\mathbf{W}_{res} in cyclic reservoir
Connection	Random connections	Ring structure
Design primitive	Randomly generated	Deterministic design
Important parameter	Spectral radius	\mathbf{W}_{res} scaling
Representation	Matrix with random number	Shifted identity matrix
Physical counterpart	Not available yet	Rotor (proposed in this work)

Changes in the manuscript:

Page 14, Line 338: **More discussions and comparison on the power efficiency of eRNR can be found in Supplementary Note 3.**

Supplementary Note 3. Why eRNR can be more power efficient?

From a fundamental perspective, the different mechanisms of introducing memory in rotation-based architecture and other architectures largely determine their power efficiency. In the rotation-based architecture, the memory is provided by the rotating dynamic node itself (see **Fig. 3a** and **Methods**). The excellent consistency between the rotation behavior and software algorithm frees the system from using extra control units, ADC and memory, which remarkably reduce the system complexity and power consumption. Also, implementing the logic switches for rotation is a resource-efficient approach by using CMOS-based transmission gates. Meanwhile, the rotating dynamic node serves to process signal and retain previous information simultaneously. Such in-memory computing paradigm is advantageous for low-power computing. In other architectures, such as the well-studied delay-based one, the memory is actually separated from the processor. Although the processing carried out in nonlinear dynamic node was a significant progress, the memory is mainly provided by the delay unit which is constrained by the limitations of conventional digital computing, such as power consumption, throughput and latency¹³. These fundamental differences result in the better power efficiency for the rotation-based architecture.

Compared with the classic random reservoir computing, the key difference of cyclic reservoir is the connection in the reservoir layer defined by \mathbf{W}_{res} . The \mathbf{W}_{res} of random reservoir is a randomly generated matrix with a proper spectral radius, while the cyclic counterpart is a shifted identity matrix which can be implemented in a more deterministic manner without performance degradation¹¹. In this work, it has been proven that the cyclic \mathbf{W}_{res} can be equivalent to a physical rotor (see **Methods**), while an effective physical counterpart of random \mathbf{W}_{res} is yet to be found, which remains an exciting challenge to be addressed for future studies.

Comment #2:

Suitable mask matrix can enhance the nonlinear dynamics in the reservoir. What is the influence of mask matrix (or the input layer matrix) on the proposed RNR? Will the performance be enhanced if chosen a non-binary mask matrix?

Response:

Thank you for raising this important point. There is actually an earlier study that compared the binary \mathbf{W}_{in} and multilevel \mathbf{W}_{in} (see **Fig. S2**).³¹ Interestingly, the result suggested that there is no obvious difference between the two different \mathbf{W}_{in} configurations. In response to your comments, we have also carried out additional simulations to compare the NARMA10 results of using binary weights and uniform distribution of multilevel weights for cyclic reservoir. Other parameters in Eq. (3) are the same: the standard deviation of \mathbf{W}_{in} is 0.5 (tuned by α), network size is 400, and β is 0.75.

$$s(k + 1) = f[\alpha W_{in}u(k + 1) + \beta W_{res}s(k)] \quad (3)$$

The results are: NRMSE of multilevel weights is 0.2116 ± 0.0146 , NRMSE of binary weights is 0.2177 ± 0.0117 , as shown in **Fig. S3**. The multilevel weights yielded almost the same result as binary weights. In practice, **for the convenience of hardware implementation**, the reconfigurable input layer using binary weights can be easily realized by shifting the negative and positive signal sources, as shown in **Fig. 2a** in the main text. This is also a key design allowing the system to directly interface with analog sensory signals. In comparison, for the multilevel weights, it would be much more complicated to be implemented since additional memory would be required. We have clarified this important point in the revision.

Figure S2. Impact of different input masks. [Adopted from Kuriki, Y. et. al., 2018³¹]

Figure S3. Comparison of NARMA10 result with multilevel and binary mask matrices.

Changes in the manuscript:

Page 5, Line 106: The implementation of the input layer using binary weights is also a key design allowing the system to directly interface with analog sensory signals. W_{in}

can be a matrix consisting of a randomly generated uniform distribution of -1 and 1, which has been proved as effective as multilevel weights³¹.

Reference added:

26 Kuriki, Y., Nakayama, J., Takano, K. & Uchida, A. Impact of input mask signals on delay-based photonic reservoir computing with semiconductor lasers. *Opt. Express* **26**, 5777-5788, (2018).

Comment #3:

Could the authors discuss more about the impact of the consistency between each nonlinear node on the system performance?

Response:

We thank the reviewer for this comment. As the reviewer pointed out, the hardware consistency (or variability) of each nonlinear node could affect the system performance. In our eRNR design, the nonlinear nodes are implemented by standard resistors, capacitors and diodes (**Fig. 2c**), which have quite small and negligible device variations. This variability problem would become more prominent when novel devices or materials (such as dynamic memristors or spintronic devices) are used in such computing methods. According to previous studies, an interesting and important conclusion is that a certain degree of device variation would be beneficial to the performance of physical reservoir computing by enhancing the state richness^{2,8}. In practice, how to precisely control the device variability to improve system performance remains challenging and warrants future explorations.

Changes in the manuscript:

Page 19, Line 433: The dynamic node working in a physical reservoir may suffer from device variation and impact the system performance. Previous studies find that a certain degree of device variation may be beneficial to the system performance by enhancing the state richness^{2,8}, where how to precisely control the device variability warrants future explorations.

We would like to thank the reviewers again for taking the time to assess our manuscript.

Reference

- 1 Jaeger, H. Adaptive nonlinear system identification with echo state networks. *Advances in neural information processing systems* **15**, 609-616 (2002).
- 2 Moon, J. *et al.* Temporal data classification and forecasting using a memristor-based reservoir computing system. *Nat. Electron.* **2**, 480-487, doi:10.1038/s41928-019-0313-3 (2019).
- 3 Du, C. *et al.* Reservoir computing using dynamic memristors for temporal information processing. *Nat. Commun.* **8**, 2204, doi:10.1038/s41467-017-02337-y (2017).
- 4 Alomar, M. L. *et al.* Efficient parallel implementation of reservoir computing systems. *Neural Computing and Applications* **32**, 2299-2313, doi:10.1007/s00521-018-3912-4 (2020).
- 5 Kleyko, D., Frady, E. P., Kheffache, M. & Osipov, E. Integer Echo State Networks: Efficient Reservoir Computing for Digital Hardware. *IEEE Trans. Neural Networks Learn. Syst.*, 1-14, doi:10.1109/TNNLS.2020.3043309 (2020).
- 6 Alomar, M. L. *et al.* Digital Implementation of a Single Dynamical Node Reservoir Computer. *IEEE Trans. Circuits Syst. II Express Briefs* **62**, 977-981, doi:10.1109/TCSII.2015.2458071 (2015).
- 7 Brunner, D., Soriano, M. C., Mirasso, C. R. & Fischer, I. Parallel photonic information processing at gigabyte per second data rates using transient states. *Nat. Commun.* **4**, 1-7 (2013).
- 8 Zhong, Y. *et al.* Dynamic memristor-based reservoir computing for high-efficiency temporal signal processing. *Nat. Commun.* **12**, 408, doi:10.1038/s41467-020-20692-1 (2021).
- 9 Torrejon, J. *et al.* Neuromorphic computing with nanoscale spintronic oscillators. *Nature* **547**, 428, doi:10.1038/nature23011 (2017).
- 10 Appeltant, L. *et al.* Information processing using a single dynamical node as complex system. *Nat. Commun.* **2**, 468, doi:10.1038/ncomms1476 (2011).
- 11 Rodan, A. & Tino, P. Minimum Complexity Echo State Network. *IEEE Trans. Neural Networks* **22**, 131-144, doi:10.1109/TNN.2010.2089641 (2011).
- 12 Tanaka, G. *et al.* Recent advances in physical reservoir computing: A review. *Neural Networks* **115**, 100-123, doi:<https://doi.org/10.1016/j.neunet.2019.03.005> (2019).
- 13 Indiveri, G. & Liu, S. Memory and Information Processing in Neuromorphic Systems. *Proc. IEEE* **103**, 1379-1397, doi:10.1109/JPROC.2015.2444094 (2015).
- 14 Kendall, J. D. & Kumar, S. The building blocks of a brain-inspired computer. *Appl. Phys. Rev.* **7**, 011305, doi:10.1063/1.5129306 (2020).
- 15 Soriano, M. C. *et al.* Delay-Based Reservoir Computing: Noise Effects in a Combined Analog and Digital Implementation. *IEEE Trans. Neural Networks Learn. Syst.* **26**, 388-393, doi:10.1109/TNNLS.2014.2311855 (2015).
- 16 Lilak, S. *et al.* Spoken Digit Classification by In-Materio Reservoir Computing With Neuromorphic Atomic Switch Networks. *Frontiers in Nanotechnology* **3**, 38, doi:10.3389/fnano.2021.675792 (2021).

- 17 Nakajima, K. *et al.* A soft body as a reservoir: case studies in a dynamic model of octopus-inspired soft robotic arm. *Front. Comput. Neurosci.* **7**, 1-19, doi:10.3389/fncom.2013.00091 (2013).
- 18 Duport, F., Smerieri, A., Akrou, A., Haelterman, M. & Massar, S. Fully analogue photonic reservoir computer. *Sci. Rep.* **6**, 22381, doi:10.1038/srep22381 (2016).
- 19 Liu, Z. *et al.* Multichannel parallel processing of neural signals in memristor arrays. *Sci. Adv.* **6**, eabc4797, doi:10.1126/sciadv.abc4797 (2020).
- 20 Liu, Z. *et al.* Neural signal analysis with memristor arrays towards high-efficiency brain-machine interfaces. *Nat. Commun.* **11**, 4234, doi:10.1038/s41467-020-18105-4 (2020).
- 21 Wu, W. *et al.* in *2018 IEEE Symposium on VLSI Technology*. 103-104 (IEEE, 2018).
- 22 Yao, P. *et al.* Fully hardware-implemented memristor convolutional neural network. *Nature* **577**, 641-646, doi:10.1038/s41586-020-1942-4 (2020).
- 23 Yu, J. *et al.* in *2021 Symposium on VLSI Technology*. 1-2 (IEEE, 2021).
- 24 Milano, G. *et al.* In materia reservoir computing with a fully memristive architecture based on self-organizing nanowire networks. *Nat. Mater.*, published online, doi:10.1038/s41563-021-01099-9 (2021).
- 25 Sangster, F. L. J. & Teer, K. Bucket-brigade electronics: new possibilities for delay, time-axis conversion, and scanning. *IEEE J. Solid-State Circuits* **4**, 131-136, doi:10.1109/JSSC.1969.1049975 (1969).
- 26 Sangster, F. Integrated bucket-brigade delay line using MOS tetrodes. *Philips Tech. Rev.* **31**, 266 (1970).
- 27 Joshi, V. *et al.* Accurate deep neural network inference using computational phase-change memory. *Nat. Commun.* **11**, 2473, doi:10.1038/s41467-020-16108-9 (2020).
- 28 Spoon, K. *et al.* Toward Software-Equivalent Accuracy on Transformer-Based Deep Neural Networks With Analog Memory Devices. *Front. Comput. Neurosci.* **15**, doi:10.3389/fncom.2021.675741 (2021).
- 29 Kariyappa, S. *et al.* Noise-Resilient DNN: Tolerating Noise in PCM-Based AI Accelerators via Noise-Aware Training. *IEEE Trans. Electron Devices* **68**, 4356-4362, doi:10.1109/TED.2021.3089987 (2021).
- 30 Romera, M. *et al.* Vowel recognition with four coupled spin-torque nanoo oscillators. *Nature* **563**, 230-234, doi:10.1038/s41586-018-0632-y (2018).
- 31 Kuriki, Y., Nakayama, J., Takano, K. & Uchida, A. Impact of input mask signals on delay-based photonic reservoir computing with semiconductor lasers. *Opt. Express* **26**, 5777-5788, doi:10.1364/OE.26.005777 (2018).

REVIEWERS' COMMENTS

Reviewer #1 (Remarks to the Author):

I am satisfied with the responses to my comments and with the changes made to the manuscript, particularly the additional results added to Supplementary Material. I have no further comments and recommend publication.

Reviewer #2 (Remarks to the Author):

Second report on
Rotating neurons for all-analog implementation of cyclic reservoir computing

I thank the authors for having carefully taken into account the comments of the reviewers. I understand now much better the significance of their work, and can recommend that it be published in Nature Communications. A few points should still be addressed.

First, the English is weak, with many mistakes and even a few sentences are misleading or incomprehensible. I list below an INCOMPLETE list of mistakes. The authors should have the manuscript reread by a native English speaker. Such mistakes will decrease the impact of their work, as it makes it more difficult for the reader to assess what has been achieved.

Second, numerical simulations of the CRC are described. Please specify whether realistic noise levels are taken into account in the simulators, or whether the simulators are noise free. This is particularly important in regards to the NARMA10 benchmark, which to my knowledge is very difficult to carry out in noisy systems. Personally I attribute the very good performance on NARMA10 to absence of noise, and to the size of the system: there seem to be 388×50 trained output weights.

Third. There is sometimes a confusion about whether power levels are estimated for the CMOS simulations, or measured for the demonstrator. For instance please specify in Supplementary Tables 1 and 2. (And it would be most interesting if you could give both numbers: for simulations and for demonstrator).

Incomplete list of English mistakes, unclear sentences, etc..

Abstract

the all-analog reservoir computing system achieved 94.0% accuracy with $>1000\times$ lower power than prior works.

Are you refereeing to the actual demonstrator you built, or to your simulations of CMOS system? This sentence suggests the first, but the main text indicates the later.

reservoir achieves record-low errors in A time-series prediction benchmark

By integrating A memristor array as A fully-connected output layer

Last sentence of abstract: unclear. Please rephrase.

Introduction

recurrent neural network with much lower training cost. Either lower training cost THAN something; or LOW training cost.

In principle, the complex dynamicS generated by the

Furthermore, reservoir computing is powerful FOR processing temporal signalS owing to the recurrent connections that create the dependency between the current and the past neurons dynamicS, which is also known as

(ii) in the absence of the delayed feedback line, the reservoir computing hardware suffers from the trade-off between memory capacity (MC) and state richness.

Unclear sentence. Are you sure this is what you want to say?

indicated that the eRNR system WOULD consume as low as 32.7microW for the handwriting recognition task, which WOULD BE more than three orders of magnitudes lower than literature-reported reservoir systems. These results show the tremendous potential of the proposed RNR, offering a novel paradigm FOR resource-efficient reservoir computer.

Results

it is universal that various rotating components can be considered as a reservoir computing by embedding a dynamic neuron. UNCLEAR. Particularly use of Universal.

Win IS TAKEN TO BE a matrix consisting of a randomly generated uniform distribution

bits counter, but different channel sequences connecting to the neurons. UNCLEAR.

L122 which corresponds to the OPERATION (or TRANSFORMATION) $RkWinu(k)$, as described in the Methods section,

Upon ADDING TO ITS CURRENT STATE the input of $RkWinu(k)$, the neuron output $a(k)$ is represented by the

The post-neuron rotor is a mirror operation of the input multiplexer array for implementing the transposed matrix RT . R seems to be missing an index k

Note that most of the recently proposed devices and materials for the neuron in delay-based reservoir

computing could satisfy the criteria for RNR's neuron. Not really clear what you mean.

in neuromorphic computing system, the electronics directly interacting with the environment and natural signals NEEDS TO exhibit a much longer time constant (e.g., >millisecond scale) compared with that of typical digital systems³⁰. A fast time constant could result in INSUFFICIENT MC to retain history information.

To better understand how the number of parallel RNRs affected the prediction, the states within 360s ... Unclear Sentence.

In comparison, the system performance could degrade as tau increases when predicting more steps ahead (Supplementary Fig. 1g). Unclear sentence.

processing and feature extraction, are massively required. Unclear, particularly use of MASSIVELY

system complexity and power consumption but cannot be neglected under **ARE NECESSARY IN** conventional physical RC, which remains a **AND REMAIN A KEY CHALLENGE *** key challenge for practical deployments**NO S**

AS that the processor can act as a direct sensor interface for cognitive computing purposes

To demonstrate **the** analog near-sensor computing, REMOVE THE

and this experiment DEMONSTRATES that five different handwriting vowels

Also, one important advantage of using eRNR is that its short-term memory property allows the network to retain the fading information of previous inputs in the state matrix at every time step. THIS IS GENERIC FOR RESERVOIR COMPUTING, NOT SPECIFIC TO eRNR. Maybe revise sentence.

Further advancement in this system involves the analog output weights stored in our memristor crossbar array. Not clear why memristors are mentioned here, as they are discussed later. Maybe remove this sentence.

Using the labeling, training and testing procedure introduced in the Methods section, 683 handwritings in the testing set (in total 703 handwritings) were correctly recognized, Please replace here and throughout "handwritings" by "handwritten vowels". Maybe better: "of the 703 handwritten vowels in the test set, 683 were correctly recognized"

Here the software-trained Wout was deployed onto our demonstration platform interfacing the eRNR hardware to perform real-time nearsensor handwriting recognition (see Supplementary Video 2). UNCLEAR SENTENCE

In the noise-aware training, a Gaussian white noise of ± 0.03 was added to the normalized training state data **Is this the standard deviation? Please be more precise.**

and the standard deviation (target conductance-measured conductance) was about UNCLEAR. Particularly “conductance-measured conductance” seems garbled.

system that does not need to consume energy on writing and reading binary data frequently
THAT RARELY NEEDS TO CONSUME ENERGY ON WRITING AND READING BINARY DATA.

lower power consumption compared with previous cutting-edge reservoir computing systems **in recent years**REMOVE, whose values are in the range

dynamic neuron array, laying a solid foundation for ***the following***REMOVE hardware implementation.

The overall system power consumption was estimated to be as low as 32.7 microW for the handwriting tasks operating at 10 Hz

It is unclear whether the power consumption numbers are for the demonstrator or for the CMOS simulations. Please check and clarify throughout.

most computations occur in the analog domain that only contribute**s**REMOVE S to static power

could enhance the classification performance for inputS of different classes. Also, from THE hardware perspective

Methods

feature space where different INPUT CLASSES can be linearly separated. For n OUTPUT CLASSES, only the output weights

Then, the matrix R CORRESPONDS to one-time shifting in a ring

Furthermore, the output of dynamic neurons is ** determined by both the shifted input and the previous states ** (better than subjected to both shifted input and previous states:)

the signal is fed into the dynamic nonlinear neurons and output $a(n+1)$. If **Argument of a is $a(k+1)$ or $a(n+1)$. Please check**

which remarkably reduceS the system complexity and power consumption

By observing Eq. (3), it **is suggested that** BAD FORMULATION. Maybe “It appears that” a dynamic neuron for the proposed RNR should satisfy

could be considered as a dynamic neuron under RNR architecture by coupling the time constant between the neuron and rotors.UNCLEAR

The typical values of germanium diode FOR GERMANIUM DIODES

case of linear neurons, the last TERM

THEREFORE, given a properly configured RNR, its CR counterpart

Assuming that this $a(k)$ is generated by a software CR, a comparative neuron update can be defined:
UNCLEAR

At last,REMOVE data of 1103 handwritings**HANDWRITTEN VOWELS** (2802s) were successfully collected. The 478 location and class of each handwriting**HANDWRITTEN VOWEL** were labelled at the final rising/falling edge of the X, Y raw data. We labeled the end of each handwriting event (the green square in the graph ***PLEASE SPECIFY WHICH FIGURE***) where the state matrix at this time step contains the information of **this handwriting trace**THE HANDWRITTEN TRACE because of the **effect of MC** MEMORY CAPACITY.

According to the point by-point COMPUTATION mentioned above

and the OTHER points remain 0.

considerably increase for the rotating rateS (1/ 532) below 100 kHz

SUPPLEMENTARY MATERIAL

steps ahead, which implies its **memristive** property BAD CHOICE OF WORD

for rotation is a resource-efficient **approach by using**USE OF CMOS-based transmission gates

Although the processing carried out in ALTHOUGH CARRYING OUT THE PROCESSING IN THE nonlinear dynamic node was a significant progress

Reviewer #3 (Remarks to the Author):

The authors have addressed all my comments in the revised manuscript. I recommend publication.

** See Nature Research's author and referees' website at www.nature.com/authors for information about policies, services and author benefits

Response Letter to reviewers' Comments

We sincerely appreciate the reviewers' constructive comments for helping improve our manuscript in the 1st round of revision. Also, we really appreciate Reviewer #2's close check and insightful suggestions to further improve the manuscript. Below are the point-by-point responses to each comment. All the changes to the manuscript are marked in blue.

Reviewer #1

General comments:

I am satisfied with the responses to my comments and with the changes made to the manuscript, particularly the additional results added to Supplementary Material. I have no further comments and recommend publication.

Response:

We are grateful for the reviewer's constructive comments in the 1st round of revision. We are glad that the reviewer is now satisfied with our revision and recommends our work for publication on Nature Communications.

Reviewer #2

General comments:

I thank the authors for having carefully taken into account the comments of the reviewers. I understand now much better the significance of their work, and can recommend that it be published in Nature Communications. A few points should still be addressed.

Response:

We are grateful for the reviewer's constructive comments in the 1st round of revision and recognizing the significance of our work. We have further revised our manuscript following the reviewer's suggestions. Our point-by-point responses to your comments are as follows.

Comment #1:

First, the English is weak, with many mistakes and even a few sentences are misleading or incomprehensible. I list below an INCOMPLETE list of mistakes. The authors should have the manuscript reread by a native English speaker. Such mistakes will decrease the impact of their work, as it makes it more difficult for the reader to assess what has been achieved.

Response:

We appreciate the reviewer's close check for the language problems. We have corrected all the mistakes according to the reviewer's comments. In addition, as suggested by Editor Dr. Iryna Omelchenko, the manuscript has been further edited by the qualified native English speaking editors from Springer Nature Author Service (see the certificate below). We hope the revised manuscript could be now acceptable for publication.

**Comment #2:**

*Second, numerical simulations of the CRC are described. Please specify whether realistic noise levels are taken into account in the simulators, or whether the simulators are noise free. This is particularly important in regards to the NARMA10 benchmark, which to my knowledge is very difficult to carry out in noisy systems. Personally I attribute the very good performance on NARMA10 to absence of noise, and to the size of the system: there seem to be 388*50 trained output weights.*

Response:

Thank you for your comment. Our simulator was indeed noise-free for the purpose of analyzing the working mechanism of the rotating neuron reservoir. We agree with

the reviewer that the simulation result will be worse if noise is taken into consideration. In fact, all neuromorphic computing systems that work in the analog domain inevitably suffer from noise problem. In our study, we surprisingly found that a specific region of nonlinear transformation results in a much better performance on NARMA10 system approximation, which reveals an important fact that the rich dynamics can be explored as computing resource to enhance the approximation performance. It also demonstrates the computing potentials of the proposed hardware, which is in line with the purpose of neuromorphic engineering that fully explores physical dynamics for computing.

Besides the absence of noise, we respectfully disagree with the reviewer that the performance is attributed to the large network size. In our study, the large network size of 388×50 is for the case of parallel RNRs, which achieved a NRMSE value of 0.055. For a single RNR with a smaller network size of 400×1 , the achieved NRMSE is 0.078. **Both values are record low** compared with literature reported noise-free models, as summarized in **Table R1**. For comparison, Appeltant et al. (2011) claimed that their result (NRMSE=0.15) was the best one using a hardware-based model (noise-free simulation)¹. Meanwhile, it has also been found that an earlier work² using a software-based echo state network achieved NMSE = 0.0098 (equivalent to NRMSE of 0.099). It is also found that the NRMSE value cannot keep decreasing by simply expanding the network size. It will converge at a certain level. In our opinion, the record performance achieved in this work should be mainly attributed to the hardware-based neuron dynamic and the proposed eRNR architecture, which is the key innovation of our work. To clarify this point and also stimulate follow-up studies in the community, we have revised the manuscript accordingly and published the source code of our eRNR simulator (see **Code availability**).

Table R1. Comparison of NARMA10 results

Works	Network	Implementation	Network size	NRMSE
Jaeger H. (2003)	ESN	Software	400×1	0.099
Appeltant et al. (2011)	Delay-based reservoir	Hardware model (noise-free simulation)	400×1	0.15
This work	eRNR	Hardware model (noise-free simulation)	400×1	0.078
			388×50	0.055

Changes in the manuscript:

Page 6, Line 148: Moreover, a noise-free simulator was developed to evaluate the performance of the eRNR under different configurations and demonstrate its equivalence to a CR (as proven analytically in the **Methods** section).

Page 8, Line 193: The noise-free simulation result is plotted in **Fig. 3c**

Page 9, Line 209: This result demonstrates the tremendous potential of the eRNR in high-order nonlinear system approximation due to the rich physical dynamics of electronics devices.

Comment #3:

Third. There is sometimes a confusion about whether power levels are estimated for the CMOS simulations, or measured for the demonstrator. For instance please specify in Supplementary Tables 1 and 2. (And it would be most interesting if you could give both numbers: for simulations and for demonstrator).

Response:

Thank you very much for pointing this out. The power results were estimated by the simulation of the CMOS circuit using standard 65nm technology, where an eRNR was designed and simulated. Actually, the device models and library provided by the foundry are quite accurate, so the simulation results are close to actual silicon chips. In Supplementary Table 1, the previous works that reported their system-level power were also the simulation results or roughly estimated results. We have further clarified that these are simulation results in the revised manuscript. Regarding the demonstrator implemented by discrete components, the power can be estimated by referring to their datasheets as shown in **Table R2**. Such demonstration system only serves to prove the proper functionality and signal flows of eRNR, and the estimation of its power consumption (dominated by associative parts) is not meaningful.

Table R2. Power breakdown for the eRNR demonstration system

Part	Description	Purpose	Power
eRNR	74LS161 + 128 CD4051+ Neuron circuits	Implementation of eRNR	$0.13+25.6+26.1$ $= 51.8\mu\text{W}$
Microcontroller	STM32F103	Experimental data collection and transmission	$\sim 40\text{mW}$
PC + Display (LabVIEW)	\	Data collection; Visualization; Real-time demonstration	$\sim 100\text{W}$

Changes in the manuscript:

Page 14, Line 333: The power **estimation and simulation** are described in the **Methods**, where the power of eRNR was estimated by the simulation of the CMOS circuit using foundry-provided library. The result indicates that the eRNR method can reduce the system power consumption for the handwriting task and chaotic signal prediction to 32.7 μW . The **simulation** also suggests that the static power, mainly associated with the dynamic neurons and the leakage current of transistors, plays a dominant role when the processing rate ($1/\tau_r$) is lower than 100 kHz (for which the power consumption was estimated to be 79.1 μW).

Page 23, Line 549: The **simulated** power breakdown at different frequencies is shown in Supplementary Table 2.

Supplementary Table 2: **Simulated** power breakdown for 8×8 eRNR system (μW)

Page 15, Line 375: **In the simulation of the eRNR circuit**, the overall system power consumption was estimated to be as low as 32.7 μW for the handwriting tasks operating at 10 Hz ($\tau_r = 0.1$ s), reflecting an advantage of more than three orders of magnitude compared to the consumption reported for reservoir computing systems in the literature.

Minor comments:

Abstract

the all-analog reservoir computing system achieved 94.0% accuracy with >1000× lower power than prior works.

Are you refereeing to the actual demonstrator you built, or to your simulations of CMOS system? This sentence suggests the first, but the main text indicates the later.

Response:

Here we are referring to the simulation result. We have revised this sentence as follows.

Changes in the manuscript:

By integrating a memristor array as a fully-connected output layer, the all-analog reservoir computing system **achieves** 94.0% accuracy, **while simulation shows >1000× lower system-level power than prior works.**

*reservoir achieveS record-low errors in A time-series prediction benchmark
By integrating A memristor array as A fully-connected output layer*

Response:

We have corrected those grammatic errors as you pointed out. Thank you.

Last sentence of abstract: unclear. Please rephrase.

Response:

We have revised the sentence as you suggested. Thank you.

Changes in the manuscript:

Page 2, Line 35: Therefore, our work demonstrates an elegant rotation-based architecture that explores hardware physics as computational resources for high-performance reservoir computing.

Introduction

recurrent neural network with much lower training cost. Either lower training cost THAN something; or LOW training cost.

In principle, the complex dynamicS generated by the

Furthermore, reservoir computing is powerful FOR processing temporal signalS owing to the

recurrent connections that create the dependency between the current and the past neurons dynamicS, which is also known as

Response:

We have corrected those grammatic errors as you pointed out. Thank you.

(ii) in the absence of the delayed feedback line, the reservoir computing hardware suffers from the trade-off between memory capacity (MC) and state richness. Unclear sentence. Are you sure this is what you want to say?

Response:

We have revised the sentence as you suggested. Thank you.

Changes in the manuscript:

Page 3, Line 64: (ii) in the absence of the delayed feedback line, the reservoir computing system cannot simultaneously maintain proper memory capacity (MC) and state richness. For example, previous research revealed that shortening the steps of time multiplexing could improve the MC but at the cost of reducing the state richness, or vice versa³.

indicated that the eRNR system WOULD consume as low as 32.7microW for the handwriting recognition task, which WOULD BE more than three orders of magnitudes lower than literature-reported reservoir systems. These results show the tremendous potential of the proposed RNR, offering a novel paradigm FOR resource-efficient reservoir computer.

Response:

We have corrected those grammatic errors as you pointed out. Thank you.

Results

it is universal that various rotating components can be considered as a reservoir computing by embedding a dynamic neuron. UNCLEAR. Particularly use of Universal.

Response:

We intended to say that the implementation is not limited to the CMOS. Various rotating components can be developed as a reservoir according to the RNR theory. Here we have revised this sentence.

Changes in the manuscript:

Page 5, Line 105: We shall mention that the fundamental of RNR is widely applicable to various rotating components, not limited to CMOS implementations, that can be developed as a reservoir by embedding dynamic neurons.

Page 16, Line 396: It is also worth mentioning that various rotational hardware could be explored for constructing efficient pre- and post-neuron rotors, which are the key to implement the RNR.

Win IS TAKEN TO BE a matrix consisting of a randomly generated uniform distribution

bits counter, but different channel sequences connecting to the neurons. UNCLEAR.

L122 which corresponds to the OPERATION (or TRANSFORMATION) $RkWinu(k)$, as described in the Methods section,

Response:

We have corrected those grammatic errors as you pointed out. Thank you.

Upon ADDING TO ITS CURRENT STATE the input of $RkWinu(k)$, the neuron output $a(k)$ is represented by the

Response:

Thank you for the suggestion. We have revised this sentence to make it clearer.

Changes in the manuscript:

Page 5, Line 127: Upon receiving the neuron input $\gamma \mathbf{R}^k \mathbf{W}_{in} \mathbf{u}(k)$ and adding to its current value, the resulting neuron output $\mathbf{a}(k)$ is represented by the voltage level measured at the right side of the neuron circuit.

The post-neuron rotor is a mirror operation of the input multiplexer array for implementing the transposed matrix \mathbf{R}^T . \mathbf{R} seems to be missing an index k

Response:

Thank you for the suggestion. \mathbf{R} is the rotation matrix. The matrix multiplied by \mathbf{R} means a one-time shift. \mathbf{R}^k denotes k -time shifting and $(\mathbf{R}^k)^T$ denotes k -time reserved shifting. Here we intend to explain the relation between the post-neuron rotor and the transposed matrix \mathbf{R}^T regardless of the shifting times. Therefore, we think there is no need to add an index 'k' here. Thank you for your close check.

Note that most of the recently proposed devices and materials for the neuron in delay-based reservoir computing could satisfy the criteria for RNR's neuron. Not really clear what you mean.

Response:

Thank you for this comment. In recent years, various devices and materials have been used as the dynamic neuron of physical reservoir computers. After introducing the general model (**Fig. 2b**) and design details (**Methods**) of the dynamic neuron in RNR, we intend to claim that most of the recently reported devices and materials for physical reservoir computing could also be used as the neuron in the RNR architecture. The readers can link this new architecture to their previous readings and research for more innovative ideas. Here we have rephrased the sentence to make it clear.

Changes in the manuscript:

Page 6, Line 140: As discussed in **Fig. 2b** and **Methods**, most of the recently reported devices and materials for physical reservoir computing could also be used as the neuron in the RNR architecture³⁻⁵

in neuromorphic computing system, the electronics directly interacting with the environment and natural signals NEEDS TO exhibit a much longer time constant

(e.g., >millisecond scale) compared with that of typical digital systems³⁰. A fast time constant could result in INSUFFICIENT MC to retain history information.

Response:

We have corrected those grammatic errors as you pointed out. Thank you.

To better understand how the number of parallel RNRs affected the prediction, the states within 360s ... Unclear Sentence.

Response:

We have revised the sentence to make it clearer. Thank you.

Changes in the manuscript:

Page 10, Line 239: To better understand how the number of parallel RNRs (i.e., M) affected the prediction performance of the system, the states within 360 s (2880×64 samples, half for training and half for testing) were collected with the platform.

In comparison, the system performance could degrade as tau increases when predicting more steps ahead (Supplementary Fig. 1g). Unclear sentence.

Response:

We have revised the sentence to make it clearer. Thank you.

Changes in the manuscript:

Page 11, Line 254: In comparison, the system performance could degrade as τ increases for multistep ahead prediction (Supplementary Fig. 1g).

processing and feature extraction, are massively required. Unclear, particularly use of MASSIVELY

Response:

Thank you for your comment. In most applications, the received sensory signals usually need a series of operations, such as analog-to-digital conversion, pre-processing and feature extraction, before feeding them to a machine learning model/system. These operations could incur extra system complexity and power consumption, which are not desirable in low-power edge devices. Here we have replaced the word ‘massively’ with “also often”.

*system complexity and power consumption but cannot be neglected under ****ARE NECESSARY IN**** conventional physical RC, which remains a ****AND REMAIN A KEY CHALLENGE ***** key challenge for practical deployments****NO S*****

AS that the processor can act as a direct sensor interface for cognitive computing purposes

*To demonstrate ****the**** analog near-sensor computing, REMOVE THE*

and this experiment DEMONSTRATES that five different handwriting vowels

Also, one important advantage of using eRNR is that its short-term memory property allows the network to retain the fading information of previous inputs in the state matrix at every time step. THIS IS GENERIC FOR RESERVOIR COMPUTING, NOT SPECIFIC TO eRNR. Maybe revise sentence.

Response:

We have revised these sentences as you suggested. Thank you.

Further advancement in this system involves the analog output weights stored in our memristor crossbar array. Not clear why memristors are mentioned here, as they are discussed later. Maybe remove this sentence.

Response:

Thank you for your comment. In this paragraph, we intend to give an overall introduction about the all-analog system, referring to **Fig. 5b** where the memristor array-based output layer is also included. We have slightly modified the sentence as follows.

Changes in the manuscript:

Page 12, Line 288: Further advancement of this system involves the analog output weights stored in a memristor crossbar array to realize all-analog signal processing,

Using the labeling, training and testing procedure introduced in the Methods section, 683 handwritings in the testing set (in total 703 handwritings) were correctly recognized, Please replace here and throughout “handwritings” by “handwritten vowels”. Maybe better: “of the 703 handwritten vowels in the test set, 683 were correctly recognized”

Here the software-trained Wout was deployed onto our demonstration platform interfacing the eRNR hardware to perform real-time nearsensor handwriting recognition (see Supplementary Video 2). UNCLEAR SENTENCE

*In the noise-aware training, a Gaussian white noise of ± 0.03 was added to the normalized training state data ****Is this the standard deviation? Please be more precise.*****

and the standard deviation (target conductance-measured conductance) was about UNCLEAR. Particularly “conductance-measured conductance” seems garbled.

Response:

We have revised these sentences as you suggested. Thank you.

system that does not need to consume energy on writing and reading binary data frequently

THAT RARELY NEEDS TO CONSUME ENERGY ON WRITING AND READING BINARY DATA.

Response:

Thank you for your comment. Unlike conventional digital computers, the proposed all-analog implementation does not need to convert the analog signal into the digital domain for calculation. As illustrated in Fig. 2 and Fig. 5b, no digital memory, ADC and DAC is involved to read and write binary data. Instead, **all the computing is performed in the analog domain**, which saves the energy for frequent data conversion between digital and analog domains. We have modified the sentence to be more accurate.

Changes in the manuscript:

Page 14, Line 339: This striking advantage is associated with the unique all-analog computing capability of our eRNR-implemented reservoir computing system, **which saves the energy for frequent data conversion between digital and analog domains.**

*lower power consumption compared with previous cutting-edge reservoir computing systems ****in recent years**REMOVE**, whose values are in the range*

*dynamic neuron array, laying a solid foundation for *****the following***REMOVE** hardware implementation.*

The overall system power consumption was estimated to be as low as 32.7 microW for the handwriting tasks operating at 10 Hz

It is unclear whether the power consumption numbers are for the demonstrator or for the CMOS simulations. Please check and clarify throughout.

most computations occur in the analog domain that only contribute ~~s~~ to static power

could enhance the classification performance for inputs of different classes. Also, from THE hardware perspective

Methods

feature space where different INPUT CLASSES can be linearly separated. For n OUTPUT CLASSES, only the output weights

Then, the matrix R CORRESPONDS to one-time shifting in a ring

Furthermore, the output of dynamic neurons is determined by both the shifted input and the previous states (better than subjected to both shifted input and previous states:)

the signal is fed into the dynamic nonlinear neurons and output $a(n+1)$. If Argument of a is $a(k+1)$ or $a(n+1)$. Please check

which remarkably reduceS the system complexity and power consumption

By observing Eq. (3), it is suggested that BAD FORMULATION. Maybe “It appears that” a dynamic neuron for the proposed RNR should satisfy

Response:

We revised the above-mentioned sentences as you suggested. Thank you.

could be considered as a dynamic neuron under RNR architecture by coupling the time constant between the neuron and rotors. UNCLEAR

Response:

Thank you for this comment. In this **Methods** section, we start by introducing the general model of the dynamic neuron working in the proposed RNR architecture,

corresponding to **Fig. 2b**. After introducing the general model, we intend to claim in this sentence that any passive element that exhibits these three characteristics could work as a dynamic neuron in the RNR architecture. Here we have revised this sentence to make it clearer.

Changes in the manuscript:

Page 19, Line 446: Any passive element that exhibits these three characteristics could essentially be used as a dynamic neuron in the RNR architecture by fine-tuning the time constants of neuron and rotors.

The typical values of germanium diode FOR GERMANIUM DIODES

case of linear neurons, the last TERM

THEREFORE, given a properly configured RNR, its CR counterpart

Assuming that this $a(k)$ is generated by a software CR, a comparative neuron update can be defined: UNCLEAR

***At last,**REMOVE data of 1103 handwritings**HANDWRITTEN VOWELS** (2802s) were successfully collected. The 478 location and class of each handwriting**HANDWRITTEN VOWEL** were labelled at the final rising/falling edge of the X, Y raw data. We labeled the end of each handwriting event (the green square in the graph ***PLEASE SPECIFY WHICH FIGURE**) where the state matrix at this time step contains the information of **this handwriting trace**THE HANDWRITTEN TRACE because of the **effect of MC** MEMORY CAPACITY.*

According to the point by-point COMPUTATION mentioned above

and the OTHER points remain 0.

considerably increase for the rotating rateS (1/ 532) below 100 kHz

SUPPLEMENTARY MATERIAL

*steps ahead, which implies its **memristive** property BAD CHOICE OF WORD*

*for rotation is a resource-efficient **approach by using**USE OF CMOS-based transmission gates*

***Although the processing carried out in** ALTHOUGH CARRYING OUT THE PROCESSING IN THE nonlinear dynamic node was a significant progress*

Response:

We revised the above-mentioned sentences as you suggested. Thank you for your close check and valuable comments.

Reviewer #3

General comments:

The authors have addressed all my comments in the revised manuscript. I recommend publication.

Response:

We sincerely thank you for your constructive comments in the 1st round of revision. We are glad that we have addressed all your comments, and sincerely appreciate your recommendation.

We would like to thank all the reviewers again for taking the time to assess our manuscript.

Reference

- 1 Appeltant, L. *et al.* Information processing using a single dynamical node as complex system. *Nat. Commun.* **2**, 468, doi:10.1038/ncomms1476 (2011).
- 2 Jaeger, H. Adaptive nonlinear system identification with echo state networks. *Advances in neural information processing systems* **15**, 609-616 (2002).
- 3 Zhong, Y. *et al.* Dynamic memristor-based reservoir computing for high-efficiency temporal signal processing. *Nat. Commun.* **12**, 408, doi:10.1038/s41467-020-20692-1 (2021).
- 4 Torrejon, J. *et al.* Neuromorphic computing with nanoscale spintronic oscillators. *Nature* **547**, 428, doi:10.1038/nature23011 (2017).
- 5 Moon, J. *et al.* Temporal data classification and forecasting using a memristor-based reservoir computing system. *Nat. Electron.* **2**, 480-487, doi:10.1038/s41928-019-0313-3 (2019).